# Unique trajectory of gene family evolution from genomic analysis of nearly all known species in an ancient yeast lineage

Bo Feng [1,2,3,17], Yonglin Li [4,17], Biyang Xu [1,17], Hongyue Liu [1], Jacob L Steenwyk [5], Kyle T David [6,7], Xiaolin Tian [1], Carla Gonçalves [6,7,8,9], Dana A Opulente [10,11], Abigail L LaBella [6,7,12,13], Marie-Claire Harrison [6,7], John F Wolters [10], Shengyuan Shao [1], Zhaohao Chen [1], Kaitlin J Fisher [10,14], Marizeth Groenewald [15], Chris Todd Hittinger [10], Xing-Xing Shen [16], Shengying Li [2], Antonis Rokas [6,7✉], Xiaofan Zhou [4✉] & Yuanning Li [1,2,3✉]

## Abstract

Gene gains and losses are a major driver of genome evolution; their precise characterization can provide insights into the origin and diversification of major lineages. Here, we examined gene family evolution of 1154 genomes from nearly all known species in the medically and technologically important yeast subphylum Saccharomycotina. We found that yeast gene family evolution differs from that of plants, animals, and filamentous ascomycetes, and is characterized by smaller overall gene numbers yet larger gene family sizes for a given gene number. Faster-evolving lineages (FELs) in yeasts experienced significantly higher rates of gene losses—commensurate with a narrowing of metabolic niche breadth—but higher speciation rates than their slower-evolving sister lineages (SELs). Gene families most often lost are those involved in mRNA splicing, carbohydrate metabolism, and cell division and are likely associated with intron loss, metabolic breadth, and non-canonical cell cycle processes. Our results highlight the significant role of gene family contractions in the evolution of yeast metabolism, genome function, and speciation, and suggest that gene family evolutionary trajectories have differed markedly across major eukaryotic lineages.

**Keywords** Yeast; Fungi; Gene Family Evolution; Comparative Genomics
**Subject Categories** Chromatin, Transcription & Genomics; Evolution & Ecology

## Introduction

Gene duplications and losses are one of the major drivers of genome evolution and the source of major evolutionary innovations. For example, the evolutionary transition to vascular plants, originating from the common ancestor of Viridiplantae, was characterized by significant gene family expansion events, reflecting adaptations to life in terrestrial environments (One Thousand Plant Transcriptomes Initiative, 2019). Similarly, the evolution of animals was marked by the accumulation of genes essential for multicellularity (Ocaña-Pallarès et al, 2022). In contrast, the ancestors of fungi primarily experienced a reduction in most functional gene categories, with early fungal evolution featuring both the loss of ancient protist gene families and the expansion of novel fungal gene families (Merényi et al, 2023). For instance, filamentous ascomycetes, a major group within Ascomycota, underwent significant expansions in gene families associated with cytochrome P450 monooxygenases, enabling ecological adaptation (Deng et al, 2007). These distinct evolutionary trajectories underscore the diversity and adaptive strategies of eukaryotes.

[1]Institute of Marine Science and Technology, Shandong University, Qingdao 266237, China. [2]State Key Laboratory of Microbial Technology, Shandong University, Qingdao 266237, China. [3]Laboratory for Marine Biology and Biotechnology, Qingdao Marine Science and Technology Center, Qingdao 266237, China. [4]Guangdong Laboratory for Lingnan Modern Agriculture, Guangdong Province Key Laboratory of Microbial Signals and Disease Control, Integrative Microbiology Research Center, South China Agricultural University, Guangzhou 510642, China. [5]Howards Hughes Medical Institute and the Department of Molecular and Cell Biology, University of California, Berkeley, Berkeley, CA, USA. [6]Department of Biological Sciences, Vanderbilt University, Nashville, TN 37235, USA. [7]Evolutionary Studies Initiative, Vanderbilt University, Nashville, TN 37235, USA. [8]Associate Laboratory i4HB-Institute for Health and Bioeconomy and UCIBIO-Applied Molecular Biosciences Unit, Department of Life Sciences, NOVA School of Science and Technology, Universidade Nova de Lisboa, Caparica, Portugal. [9]UCIBIO-i4HB, Departamento de Ciências da Vida, Faculdade de Ciências e Tenologia, Universidade Nova de Lisboa, Caparica, Portugal. [10]Laboratory of Genetics, J. F. Crow Institute for the Study of Evolution, Center for Genomic Science Innovation, Department of Energy (DOE) Great Lakes Bioenergy Research Center, Wisconsin Energy Institute, University of Wisconsin-Madison, Madison, WI 53726, USA. [11]Biology Department, Villanova University, Villanova, PA 19085, USA. [12]Department of Bioinformatics and Genomics, University of North Carolina at Charlotte, North Carolina Research Campus, Kannapolis, NC 28233, USA. [13]Center for Computational Intelligence to Predict Health and Environmental Risks (CIPHER), University of North Carolina at Charlotte, Charlotte, NC 28233, USA. [14]Department of Biological Sciences, State University of New York at Oswego, Oswego, NY 13126, USA. [15]Westerdijk Fungal Biodiversity Institute, Utrecht 3584, The Netherlands. [16]Key Laboratory of Biology of Crop Pathogens and Insects of Zhejiang Province, Institute of Insect Sciences, Zhejiang University, Hangzhou 310058, China. [17]These authors contributed equally as first authors: Bo Feng, Yonglin Li, Biyang Xu. ✉E-mail: antonis.rokas@vanderbilt.edu; xiaofan_zhou@scau.edu.cn; yuanning.li@email.sdu.edu.cn

The Saccharomycotina subphylum (phylum Ascomycota, Kingdom Fungi) encompasses a diverse array of ~1200 species, including the well-known baker's yeast *Saccharomyces cerevisiae*, the opportunistic pathogen *Candida albicans*, and the industrial producer of oleochemicals *Yarrowia lipolytica* (Wang et al, 2016; Madzak, 2021). Species in the subphylum, which began diversifying approximately 400 million years ago, showcase remarkable ecological, genomic, and metabolic diversity (Gonçalves and Gonçalves, 2019; Marcet-Houben and Gabaldón, 2015; Hittinger, 2013; Boekhout et al, 2022; Opulente et al, 2018, 2024). From fermenting sugars to metabolizing urea and xenobiotic compounds, Saccharomycotina yeasts (hereafter referred to as yeasts) have evolved diverse metabolic pathways that allow them to thrive in environments as varied as fruit skins, deep-sea vents, arctic ice, and desert sands (Linder, 2019; Khan et al, 2023; Burgaud et al, 2010; Chen et al, 2018; David et al, 2024; Opulente et al, 2024). Genome-wide protein sequence divergence levels within the yeast subphylum are on par with those observed within the plant and animal kingdoms (Shen et al, 2018). However, gene family evolution in the yeast subphylum remains largely unexplored. This limitation has been primarily due to a concentration of research on a limited subset of species and the lack of comprehensive genomic data across the whole subphylum (Bendixsen et al, 2021; Libkind et al, 2020; Peris et al, 2023). Moreover, evolutionary analyses of a wide range of yeast species would facilitate better understanding of the specific genes and genetic mechanisms enabling them to thrive in various ecological niches.

Here, we leveraged the recent availability of 1154 draft genomes from 1051 yeast species—covering 95% of known species within the Saccharomycotina subphylum—to investigate how gene family evolution has shaped yeast diversity. By comparing this large dataset with other major eukaryotic lineages, we found that yeasts exhibit larger gene family sizes when their total gene counts are similar. For example, the yeast *Dipodascus armillariae* (9561 genes) has an average of 1.68 genes per family, compared with 1.35 in the heterokont alga *Micromonas pusilla* (10,238 genes). Our analyses revealed that certain yeast lineages exhibit notably different evolutionary rates, including shifts in key processes such as mRNA splicing, cell division, and carbon source utilization. Together, these findings highlight the importance of gene family dynamics and provide both broad and fine-scale insights into the tempo and mode of yeast evolutionary diversification.

# Results

## Gene family diversity is correlated with total gene content in eukaryotes

We sampled 1154 yeast genomes, 761 filamentous ascomycetous (from subphylum Pezizomycotina) genomes, 83 animal (Kingdom Metazoa) genomes, and 1178 plant (Kingdom Viridiplantae, Phylum Glaucophyta, and Phylum Rhodophyta) genomes and transcriptomes from previous studies (Shen et al, 2020; Liu et al, 2024; One Thousand Plant Transcriptomes Initiative, 2019; Opulente et al, 2024), representing every major lineage across these four groups (Dataset EV1). Using OrthoFinder, we identified 62,643 orthologous groups of genes (hereafter referred to as gene families) in yeasts, 137,783 in Pezizomycotina, 65,811 in animals,

and 52,956 in plants. To filter out species-specific or rare gene families, we excluded all gene families that were present in 10% or fewer of the taxa in each major lineage (the threshold of 10% was based on the density plot of gene family average coverage; Fig. EV1). This filtering resulted in the identification of 5551 gene families in yeasts (that collectively contain 89.88% of the genes assigned to orthogroups by OrthoFinder), 9473 in Pezizomycotina (~87.09%), 11,076 in animals (~76.68%), and 8231 in plants (~96.41%).

Examination of weighted average gene family sizes, calculated using the reciprocal of maximum observed gene family size as the weight to account for differences in gene family size, revealed distinct features of gene family content for each group. Specifically, yeasts and filamentous ascomycetes typically had smaller weighted average gene family sizes than animals and plants (Fig. 1A). However, when comparing organisms with equivalent numbers of protein-coding genes (e.g., when comparing a yeast genome with ~10,000 genes with a plant genome with ~10,000 genes), yeasts displayed similar weighted average sizes to plants and larger sizes than filamentous ascomycetes and animals (Fig. 1B).

Moreover, we found a strong positive correlation between the phylogenetic independent contrasts (PICs) of weighted average gene family size and the number of protein-coding genes (gene number). This correlation was particularly pronounced in plants (rho = 0.97), yeasts (rho = 0.82), and filamentous ascomycetes (rho = 0.88), but weaker in animals (rho = 0.62), with all *P*-values less than 0.01 (Fig. 1C; Appendix Table S1). The correlation between PICs of weighted average gene family size and genome size was weaker (Appendix Table S1). Our PIC regression showed yeasts had a steeper slope than plants, animals or filamentous ascomycetes (Fig. 1C). This indicates that yeasts tend to have larger gene family sizes as their gene number increases (Fig. 1B). This result suggests that yeasts tend to exhibit larger gene family sizes/gene number compared to animals and filamentous ascomycetes and are on par with plants, corroborating the contributions of gene duplications to yeast phenotypic diversity (Dujon and Louis, 2017; Mattenberger et al, 2017; Kang et al, 2019).

## Reduced gene family content is associated with rapid genome sequence evolution

The weighted average gene family size across 12 yeast orders (Groenewald et al, 2023) is 1.12 genes/gene family, with Alloascoideales having the highest size at 1.49 and Saccharomycodales having the lowest size at 0.82 (Figs. 2A and EV2). The average gene number and genome size across all 12 orders is 5908 genes and 13.17 Mb, respectively. Alloascoideales yeasts have the highest average gene numbers and genome sizes (8732 genes and 24.15 Mb, respectively), whereas Saccharomycodales have the smallest ones (4566 genes and 9.82 Mb, respectively).

Saccharomycodales contains the FEL in the genus *Hanseniaspora*, which is known to have experienced significant lineage-specific gene losses, especially in genes involved in the cell cycle and DNA repair, which are correlated with significantly higher evolutionary rates (Steenwyk et al, 2019). Thus, we first examined the correlation between weighted average gene family size and evolutionary rate across the 12 orders and found that it was moderate (rho = −0.65, *P* < 0.01) (Fig. EV3). We next tested whether weighted average gene family size and evolutionary rate

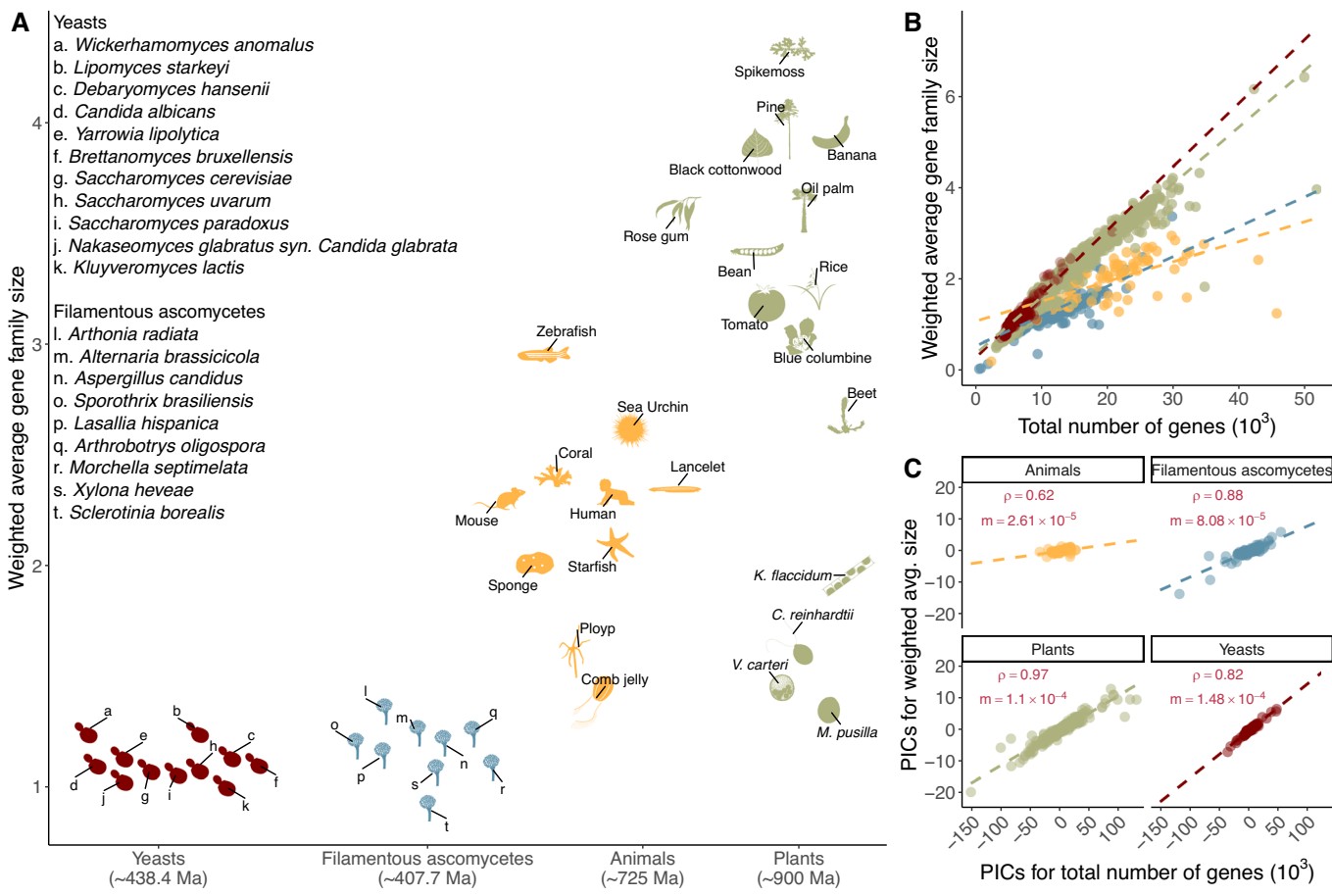

**Figure 1. Narrow range of weighted average gene family sizes among yeasts versus broader diversity in animals and plants.**

(A) The weighted average size of gene families across yeasts (from subphylum Saccharomycotina), filamentous ascomycetes (subphylum Pezizomycotina), animals (Kingdom Metazoa), and plants (Kingdom Viridiplantae, Phylum Glaucophyta, and Phylum Rhodophyta). In this study, we calculated the weighted average gene family size and performed fold change analysis using the 0.1 threshold to exclude species-specific families (Fig. EV1). All subsequent analyses were conducted using the more stringent 0.5 threshold. Representative species for yeasts and animals were identified based on previous studies (Shen et al, 2018); representatives for plants were chosen from species with available genome data; for filamentous ascomycetes, one representative per class was selected. The estimated divergence times are ~438.4 million years for yeasts, 407.7 million years for filamentous ascomycetes, 725 million years for animals, and 900 million years for plants, derived from previous studies (Shen et al, 2018, 2020; dos Reis et al, 2015; Yang et al, 2016). Images representing taxa were manually created and sourced from Phylopic (https://www.phylopic.org/). (B) Correlation plot between the weighted average gene family size and the total number of protein-coding genes across yeasts, filamentous ascomycetes, animals, and plants. (C) Correlation plot between the PICs of weighted average gene family size and the total number of protein-coding genes across yeasts, filamentous ascomycetes, animals, and plants. Correlations were determined through the Spearman test using the R package stats version 4.3.2. Specifically, the correlation coefficient (rho) for yeasts was 0.82, for filamentous ascomycetes was 0.88, for animals was 0.62, and for plants was 0.97, all statistically significant with $P < 0.01$. The slope (m) is calculated using linear regression based on the PICs of weighted average gene family size and the total number of protein-coding genes across these four groups. The PIC-related codes and data are available at the Figshare repository https://figshare.com/s/66d97c17e16c241f41e6.

varied within specific orders. We found lineage-specific variations in evolutionary rates for Dipodascales ($P = 0.04$), Saccharomycodales ($P = 0.01$), Trigonopsidales ($P < 0.01$), Pichiales ($P < 0.01$), and Serinales ($P < 0.01$) using the multimodality test (Appendix Table S2). Among the five orders that showed lineage-specific variations in evolutionary rates, only Dipodascales ($P < 0.01$), Saccharomycodales ($P = 0.02$), Trigonopsidales ($P < 0.23$, with notably fewer taxa than the remaining eight orders, all >0.75), and Pichiales (similar to Trigonopsidales) exhibited significant or notable differences in weighted average size based on the multimodality test (Appendix Table S2). The analysis of the relationship between weighted average gene family size and evolutionary rate revealed two distinct clusters within Dipodascales,

Saccharomycodales, and Trigonopsidales (Fig. 2B–D; Appendix Figs. S1 and S2). These clusters corresponded to faster-evolving lineages (FELs), characterized by smaller weighted average gene family sizes and higher evolutionary rates, and slower-evolving lineages (SELs), which exhibited larger weighted average gene family sizes and slower evolutionary rates. Specifically, differences in weighted average gene family size included median values of genes/gene family of 1.01 for FEL vs. 1.10 for SEL in Trigonopsidales, 0.93 vs. 1.17 in Dipodascales, and 0.76 vs. 0.95 in Saccharomycodales (all $P < 0.01$). For evolutionary rates, the average number of amino acid substitutions/site were 1.25 vs. 1.00 in Trigonopsidales FEL vs. SEL, 1.93 vs. 1.12 in Dipodascales FEL vs. SEL, and 2.75 vs. 1.89 in Saccharomycodales FEL vs. SEL

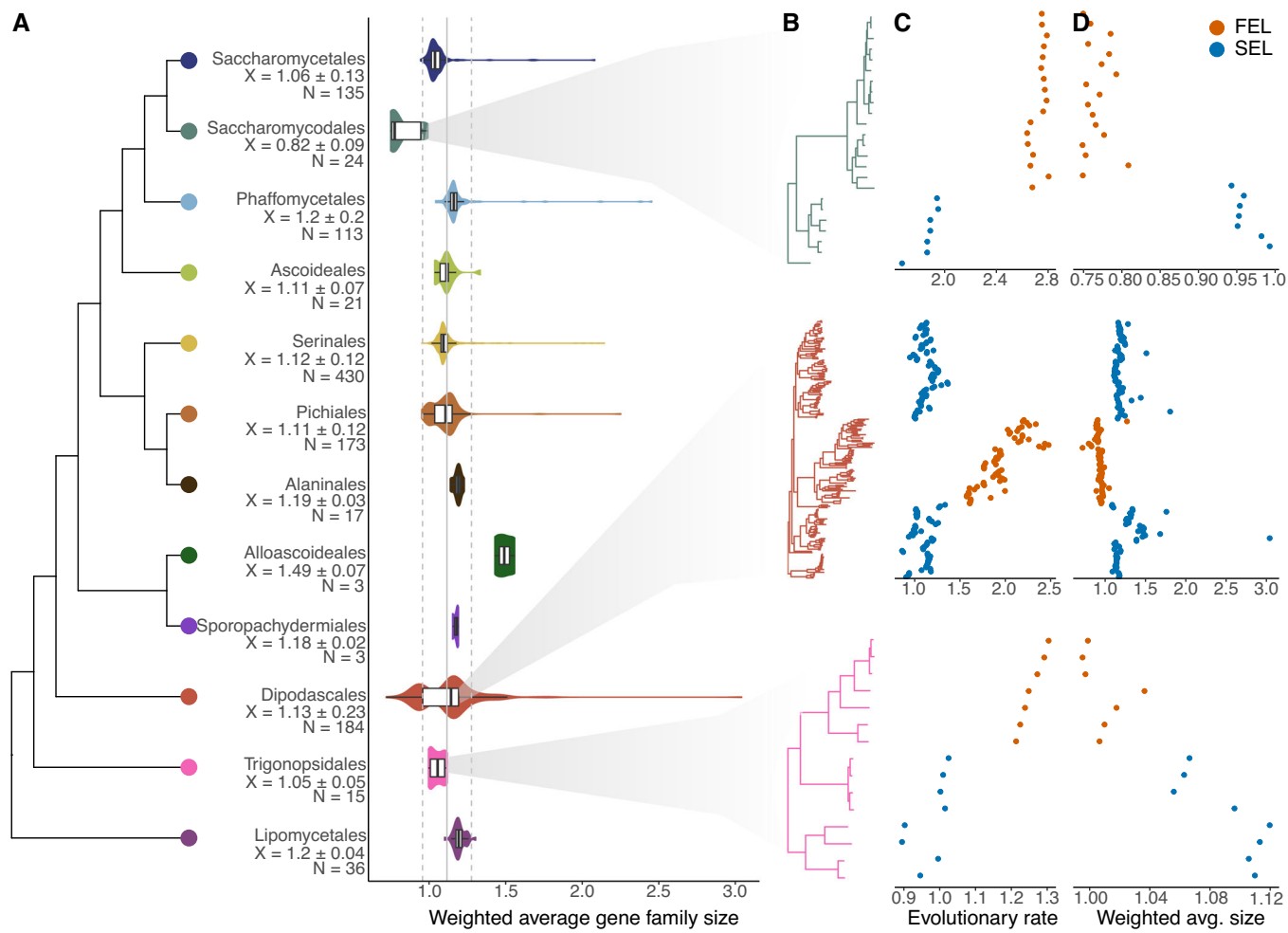

**Figure 2. Notable variations in weighted average gene family sizes within specific yeast orders.**

(A) The phylogeny of 1154 yeasts, derived from a previous study (Opulente et al, 2024). Colors indicate the taxonomic classification of species within the Saccharomycotina order. The weighted average gene family sizes (X) and genome numbers (N) for each order are displayed beneath the respective order names. The solid gray line at 1.12 represents the mean of the weighted average gene family sizes across all yeasts, and the dashed gray lines denote one standard deviation above and below the mean (±0.16). The center of each box plot represents the median performance, the box boundaries correspond to the upper and lower quartiles, and the whiskers extend to the 5th and 95th percentiles. (B) The orders Trigonopsidales, Dipodascales, and Saccharomycodales are highlighted due to their notable differences in evolutionary rates and weighted average gene family sizes. (C, D) Differences in evolutionary rates/weighted average gene family sizes within specific orders. Each dot represents a yeast in the corresponding phylogeny and is arranged according to its placement on the phylogenetic tree.

(all $P < 0.01$). Notably, all three FELs formed clades that were distinct from or emerged within SELs on the yeast phylogeny (Fig. 2B) and significantly differed in their speciation rates from SELs in two of the three lineages (DR statistic median of 0.03 vs. 0.02 in Dipodascales FEL vs. SEL, $P < 0.01$; 0.12 vs. 0.02 in Saccharomycodales FEL vs. SEL, $P < 0.01$; 0.01 vs. 0.01 in Trigonopsidales FEL vs. SEL, $P = 0.27$) (Fig. 3E).

To identify gene families with significantly different sizes between FELs and SELs, we examined the fold change in average size (non-weighted) for each gene family and for each FEL/SEL pair within the same order. Following a previous study (One Thousand Plant Transcriptomes Initiative, 2019), we categorized changes into loss events (fold change equal to 0 in FEL vs. SEL), contractions (fold change <0.67 in FEL vs. SEL), expansions (fold change >1.5 in FEL vs. SEL), and gains (fold change ~infinity in FEL vs. SEL). We found extensive and significant gene family losses and contractions

in FELs (adjusted $P \leq 0.05$) (Fig. 3A). Specifically, the fractions of gene families that experienced significant contraction or loss in FELs were 10.40% (536/5155) and 13.75% (709/5155) in Dipodascales, 3.03% (123/4056) and 15.04% (610/4056) in Saccharomycodales, and 0.89% (42/4727) and 2.54% (120/4727) in Trigonopsidales.

## Rapidly evolving lineages lost genes related to RNA splicing, cell division, and metabolism

To determine the functions of gene families contracted or lost in FELs, we performed enrichment analyses using three annotation datasets—Gene Ontology (GO) terms, InterPro annotations, and Kyoto Encyclopedia of Genes and Genomes Ortholog (KO). Functional categories enriched among gene families significantly contracted or lost in FELs relative to SELs yielded numerous GO

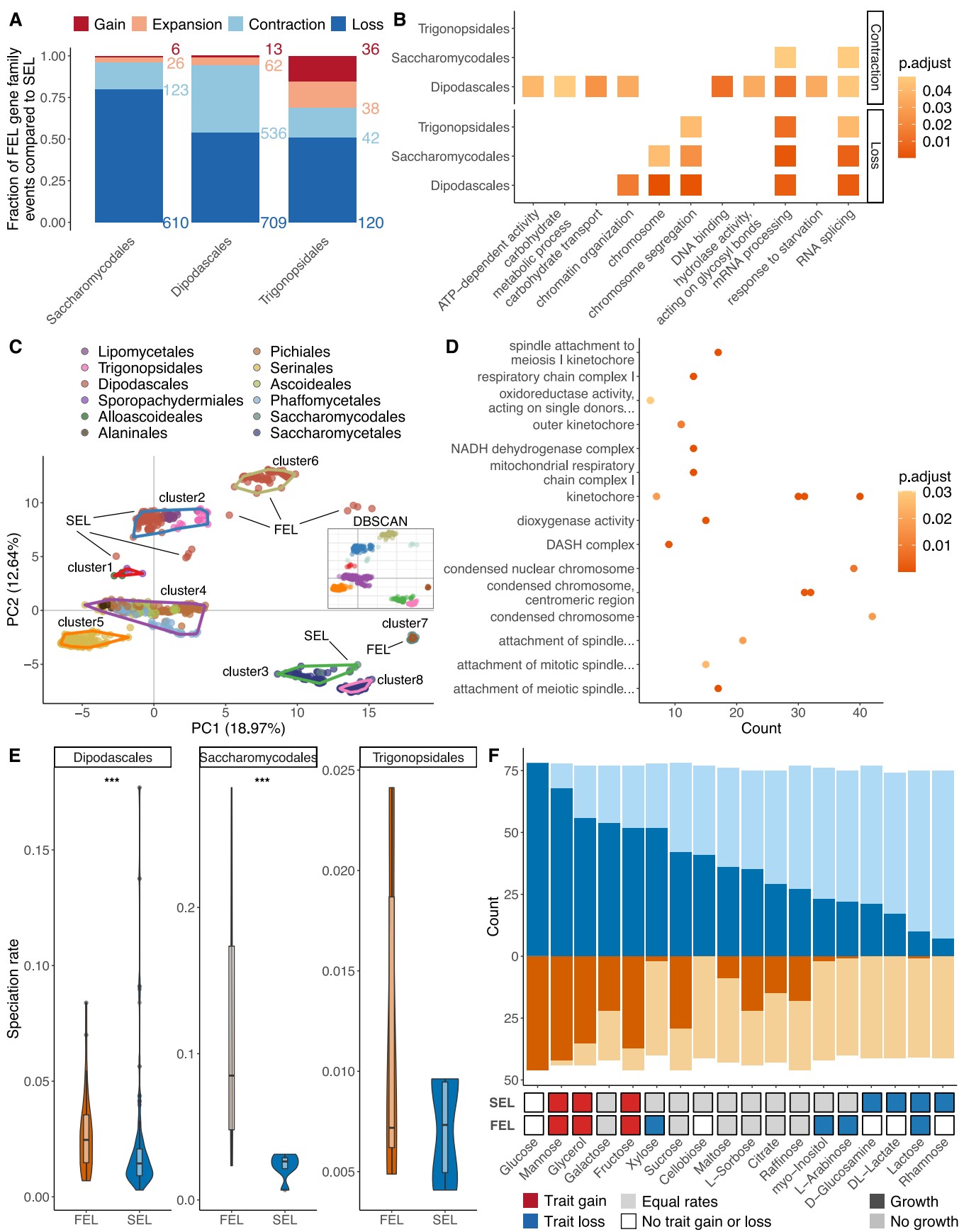

**Figure 3. Faster-evolving lineages (FELs) within three orders experienced significantly more gene family contractions and losses.**

(A) Significantly different gene family dynamics (loss, contraction, expansion, and gain) in FELs relative to SELs within Dipodascales, Saccharomycodales, and Trigonopsidales. A gene family loss is indicated by a fold change value of 0, meaning the gene family in FEL has no copies, while a fold change equal to positive infinity signifies gain. Values greater than 1.5 indicate expansion, and values less than 0.67 signify contraction. The Kolmogorov–Smirnov test was employed to assess these differences; $P \leq 0.05$. (B) GO enrichment analysis of significant contractions or losses in gene families. All enriched GO terms were simplified into GO slim terms. Fisher's exact test was used to assess significance, and p-values were adjusted using the Benjamini-Hochberg method. (C) PCA analysis utilizing presence and absence data for 4262 gene families with an average coverage of 0.5 or greater. The DBSCAN plot employs PC1 and PC2 coordinates for density-based clustering, with colors distinguishing the various clusters. In the PCA plot, points enclosed by lines indicate distinct clusters, corresponding to the color coding applied in the DBSCAN plot. (D) The GO enrichment analysis of the top 610 gene families from PC1. Fisher's exact test was used to assess significance, and p-values were adjusted using the Benjamini-Hochberg method. (E) Speciation rate comparison between FEL and SEL within Trigonopsidales, Dipodascales, and Saccharomycodales with the Wilcoxon signed-rank test, where "***" represents $P < 0.01$. The results showed statistically significant differences in Dipodascales ($P = 2.71 \times 10^{-5}$) and Saccharomycodales ($P = 4.67 \times 10^{-4}$), but not in Trigonopsidales ($P = 0.27$). The center of each box plot represents the median performance, the box boundaries correspond to the upper and lower quartiles, and the whiskers extend to the 5th and 95th percentiles. The sample sizes (n) are as follows: for Dipodascales, FEL = 61 and SEL = 123; for Saccharomycodales, FEL = 17 and SEL = 7; for Trigonopsidales, FEL = 7 and SEL = 8. (F) The evolutionary history of 17 carbon traits in FEL and SEL of Dipodascales. The dark color indicates the number of yeasts capable of utilizing the carbon source. Three different evolutionary models are shown: trait gain (red), trait loss (blue), and equal rates of trait gain and loss (gray). Estimated evolutionary models were not derived for glucose in both FEL and SEL, and for cellobiose, D-glucosamine, DL-lactate, and rhamnose in SEL, due to the uniform ability or inability of all yeasts within the group to utilize these carbon sources.

terms common across the three orders, including those associated with transcriptional functions, like RNA splicing and mRNA processing (Fig. 3B). In addition, the Dipodascales FEL experienced significant contractions in gene families related to carbohydrate metabolism. Our InterPro and Kyoto Encyclopedia of Genes and Genomes (KEGG) enrichment analyses confirmed these findings (Dataset EV2).

In addition to comparing weighted average gene family size between FELs and SELs, we illustrated the differences among yeasts based on the presence (1) and absence (0) of gene families. To exclude outliers (species-specific and/or rare gene families), we set the threshold to 0.5 based on the bimodal distribution (Fig. EV1) and carried out all subsequent analyses. A more relaxed threshold of 0.1 gave rise to highly consistent PCA distribution and correlation results (Appendix Fig. S3). Therefore, we discuss results from using the 0.5 threshold hereafter.

Following the PCA, density-based clustering according to the yeasts' position on the first two principal components (PC1 and PC2) indicated that the distributions of clusters (each corresponding to one or a few orders) generally follow the phylogeny of these orders (Figs. 2A and 3C), suggesting that patterns of gene presence or absence largely reflect yeast evolutionary relationships. Moreover, consistent with our previous findings from the fold change analysis, FELs and SELs were separated into two distinct clusters in Dipodascales (FEL in cluster 6, SEL in cluster 2) and Saccharomycodales (FEL in cluster 7, SEL in cluster 3). The FEL and SEL from Trigonopsidales were not segregated into distinct groups but were spaced apart in cluster 2. Notably, all 3 of these orders showed significant differences in the PC1 coordinates between FELs and SELs ($P \leq 0.05$) (Appendix Fig. S4).

To determine which gene families' presences or absences contribute to the distribution variation among yeasts in the PCA scatter plot (Fig. 3C), we investigated the correlation between the presence or absence of yeast gene families and their coordinates on the principal components. We identified 610 gene families whose average presence and absence in yeasts were most strongly correlated with their PC1 coordinates (rho = −0.99, $P < 0.01$), explaining significant species variation along this axis (Fig. 3C). The strong negative correlation indicates that an increase in PC1 coordinates correlates with losses in the 610 gene families, with Saccharomycodales, Saccharomycetales, and the FEL from

Dipodascales experiencing more losses than other lineages (Figs. 3C and EV4). In contrast, there was no clear relationship for gene family presence or absence along PC2 (Appendix Fig. S5). We employed the same enrichment analysis method used in the fold change analysis on these 610 gene families, revealing GO terms related to oxidoreductase activity; mitochondrial electron transport chain; and notably, cell division processes, such as the kinetochore, condensed chromosome, and DASH complex (Fig. 3D). Our InterPro and KEGG analyses echoed these findings (Dataset EV3). The enrichment results from both the fold change analysis and PCA analysis of gene presence/absence pattern (PCA analysis for short afterwards) highlighted GO terms associated with meiotic processes (adjusted $P \leq 0.05$). These include meiotic chromosome segregation (GO:0045132), kinetochore (GO:0000776), and the attachment of meiotic spindle microtubules to kinetochore (GO:0051316).

## Gene family losses suggest non-canonical spliceosomes, metabolic pathways, and DASH complexes within the FEL of dipodascales

To explore which gene families and pathways—within the enriched functional categories—experienced contraction or loss in FELs, we mapped gene families enriched in the fold change analysis and PCA analysis to the KEGG database and *Saccharomyces* Genome Database (SGD) (Wong et al, 2023), using the *S. cerevisiae* genome as a reference. Given that the FEL in Dipodascales exhibited the most significant contractions and losses of gene families compared to Saccharomycodales and Trigonopsidales, and the enrichment of RNA splicing, the DASH complex and metabolic process in fold change or PCA analyses, our study concentrated on Dipodascales. In terms of functions, we focused on the pre-mRNA splicing pathway, metabolic pathways, and the DASH complex.

The pre-mRNA splicing pathway primarily removes introns from pre-mRNA and joins exons, forming mature mRNA for protein synthesis (Wahl et al, 2009). In this pathway, 14% of the genes (12/85) exhibited contractions or losses. While *LSM8* and *PRP43* significantly contracted in the Dipodascales FEL, other gene families experienced extensive losses (Fig. 4A,B). These include *PRP40*, *CWC21*, *SNU23*, and *CWC23*, which are associated with the assembly of the spliceosomal subunits U1, U2, U4, U5, and U6

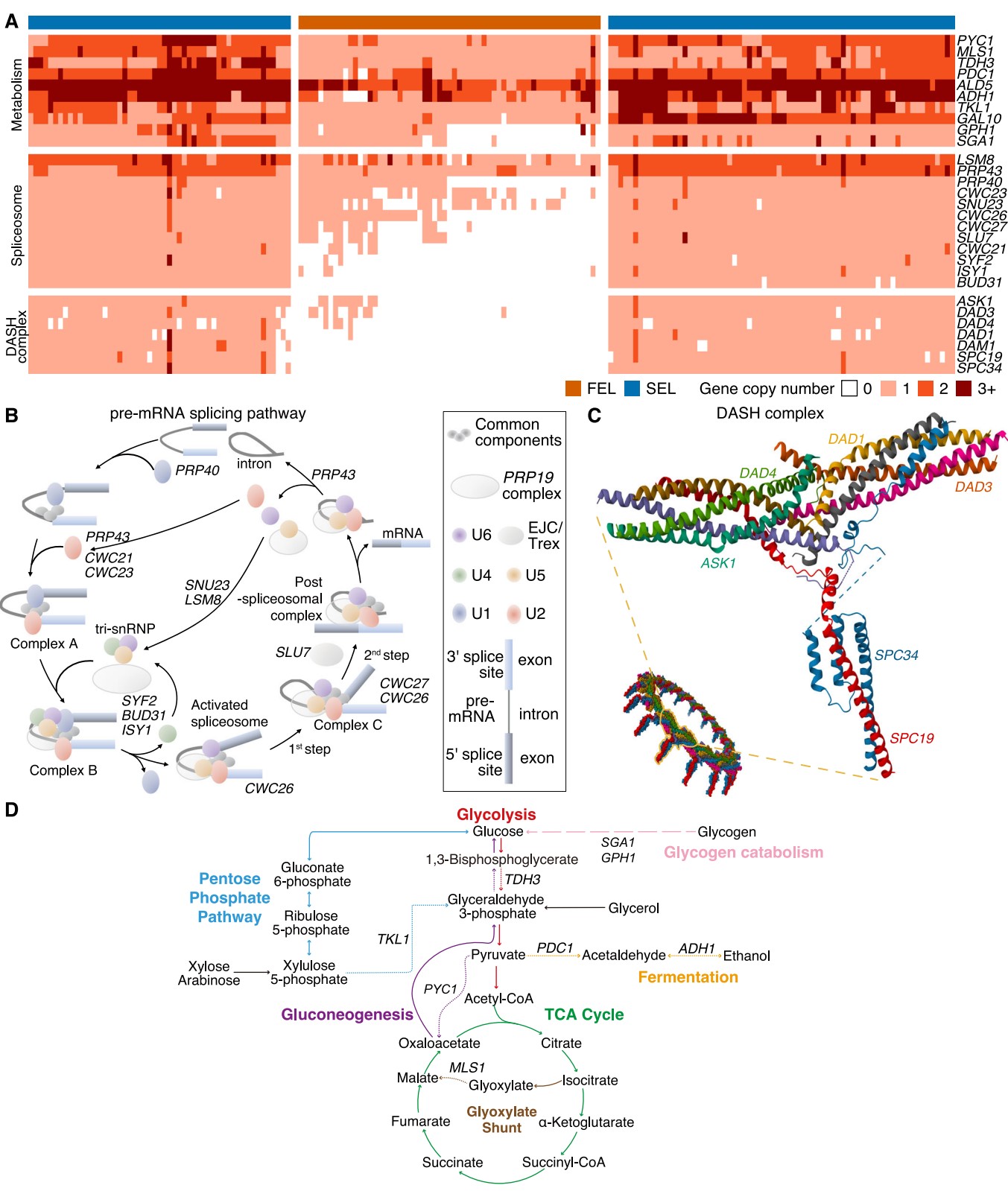

◄ **Figure 4. Dipodascales' FEL experienced the loss of key genes involved in the pre-mRNA splicing pathway, metabolic pathways, and the DASH complex.**

(A) A detailed picture of gene copy numbers in Dipodascales among metabolic pathways (10 gene families), the pre-mRNA splicing pathway (12 gene families), and the DASH complex (7 gene families). Column colors indicate SEL (blue) and FEL (orange). The estimated gene family names, identified using *S. cerevisiae* as a reference, are listed to the right of the columns. The x-axis is ordered based on the species phylogeny. (B) The pre-mRNA splicing pathway. Gene family names are marked at specific steps encoded in the pathway that experienced contractions or losses in the FEL. (C) Genes encoding the DASH complex. (D) Carbon metabolism pathways containing widespread gene loss or contraction in the Dipodascales FEL. Pathway names and reactions are indicated in corresponding colors. Steps encoded by genes experiencing contraction or loss are represented by dashed lines labeled with the gene name (gene family contractions—short dashes, gene family losses—long dashes). Pathways are abridged to show steps relevant to reported losses and contractions and not all intermediate metabolites are shown. Black arrows indicate where glycerol (gained in FEL) and xylose & arabinose (lost in FEL) feed into central carbon metabolism.

(Wahl et al, 2009). Almost all species in the Dipodascales FEL have lost genes related to the Prp19 complex, which is crucial for promoting the assembly and activation of the spliceosome, as well as stabilizing its structure (Chanarat et al, 2011). These losses could ultimately lead to abnormalities in splicing mechanisms. Notably, we found that there was significant intron loss in the Dipodascales FEL both in the total number of introns (TNI) and the average number of introns per gene (ANI) within species, with a stark reduction from a median TNI of 2815 per SEL species to 466 per FEL species ($P < 0.01$) and a decrease in ANI from 1.44 to 1.31 ($P < 0.01$) (Fig. EV5). Similar pattern of significant intron loss were observed in Trigonopsidales, with a median TNI of 6287 per SEL species vs. 789 per FEL species ($P < 0.01$) and a median ANI of 2.05 per SEL species vs. 1.31 per FEL species ($P < 0.01$) (Fig. EV5). In Saccharomycodales, the pattern was more subtle, with a median TNI of 528 per SEL species vs. 252 per FEL species ($P = 0.01$) and a median ANI of 1.22 per SEL species vs. 1.20 per FEL species ($P = 0.29$) (Fig. EV5).

The DASH complex plays a crucial role in eukaryotic cell division, particularly in chromosome segregation during mitosis (Jenni and Harrison, 2018). Strikingly, genes associated with the DASH complex were extensively lost in the Dipodascales FEL, such as *ASK1*, *DAD3*, *DAD4*, and *DAD1*, which are integral components of this complex (Fig. 4C). *DAM1*, *SPC19*, and *SPC34* were lost entirely in Dipodascales FEL species. The loss of *DAM1*, primarily involved in the stability of kinetochore microtubules, likely results in compromised microtubule stability (Westermann et al, 2006). Similarly, the absence of *SPC19* and *SPC34*, critical for the attachment of the kinetochore to microtubules, potentially leading to defects in chromosome segregation (Westermann et al, 2005).

Key metabolic pathways also exhibited considerable variation in gene family size in the Dipodascales FEL. More than half of these yeasts have lost *GPH1* and *SGA1* in the carbohydrate degradation pathway, which are responsible for encoding glycogen phosphorylase and sporulation-specific glucoamylase, respectively (Fig. 4A,D). The loss of *GPH1* and *SGA1* genes likely affects Dipodascales FEL's ability to utilize glycogen and amylopectin-like polysaccharides (Zhao et al, 2016; Wang et al, 2001). Furthermore, significant contractions were observed for *MLS1*, which encodes a key step in the glyoxylate shunt of the TCA cycle; *PYC1*, which encodes the enzyme that converts pyruvate to oxaloacetate where it can enter the TCA cycle or gluconeogenesis; *PDC1*, *ADH1*, and *ALD5*, which encode key steps in fermentation; and *TKL1*, which encodes two key reactions in the pentose phosphate pathway. We note that the present analyses reflect the known loss of the *PDC1* and *ADH1* genes in several members of the *Wickerhamiella/*

*Starmerella* (W/S) clade of the Dipodascales FEL (Gonçalves et al, 2018), but many of them reacquired alcoholic fermentation through the horizontal transfer of bacterial genes encoding alcohol dehydrogenases and the cooption of paralogs encoding decarboxylases. Further, a single FEL clade of 4 *Starmerella* species has lost *PCK1* and *FBP1*, genes essential for gluconeogenesis, *ICL1*, which encodes an essential component of the glyoxylate shunt, *GSY1*, which encodes glycogen synthase, and *GPH1* and *GDB1*, which encode the glycogen phosphorylase and glycogen debranching enzymes required for degradation of glycogen. Complete loss of *PCK1* and *FBP1* in a free-living yeast has previously been reported only in the Saccharomycodales (Steenwyk et al, 2019).

For gene families that experienced significant contractions or losses in the pre-mRNA splicing pathway, metabolic pathways, and the DASH complex in Dipodascales FEL, we observed consistent, but less pronounced, patterns in Saccharomycodales and Trigonopsidales FELs. Specifically, in the pre-mRNA splicing pathway, 50% (6/12) of genes displayed significant losses in fold change analysis in Saccharomycodales, while Trigonopsidales showed no significant changes in these genes (Appendix Table S3). All genes in Saccharomycodales had significant losses for the DASH complex, with only *DAD1* and *SPC19* similarly affected in Trigonopsidales (Appendix Table S3). No significant results were found in the metabolic pathways for genes lost in Dipodascales for either Saccharomycodales or Trigonopsidales. This outcome aligns with our enrichment results, where only a few GO terms related to these functions were enriched in Trigonopsidales, and metabolic-related functions were predominantly enriched in Dipodascales (Fig. 3B,D).

To investigate potential impacts on carbon source utilization in Dipodascales FEL, we analyzed the evolutionary trends of 18 major carbon sources (Opulente et al, 2024). We found a distinct tendency for FEL to lose growth traits associated with these carbon sources (Fig. 3F). For instance, while SEL species retained the ability to utilize cellobiose, D-glucosamine, DL-lactate, and rhamnose, FEL species have lost these growth traits. Furthermore, we found that the rate of acquiring xylose, myo-inositol, and L-arabinose growth traits in SEL species was equal to the rate of losing them. However, in FEL species, the loss rate surpassed the gain rate. Interestingly, both FEL and SEL species exhibited a greater tendency to acquire the glycerol growth trait, despite the *TDH3* gene family, which is crucial for glycerol metabolism (as well as glycolysis and gluconeogenesis), has undergone significant contraction in FEL. This result suggests the possibility of other genes or pathways being augmented to compensate for the *TDH3* contraction and enable glycerol metabolism (Klein et al, 2017). These observations suggest that gene losses and contractions in

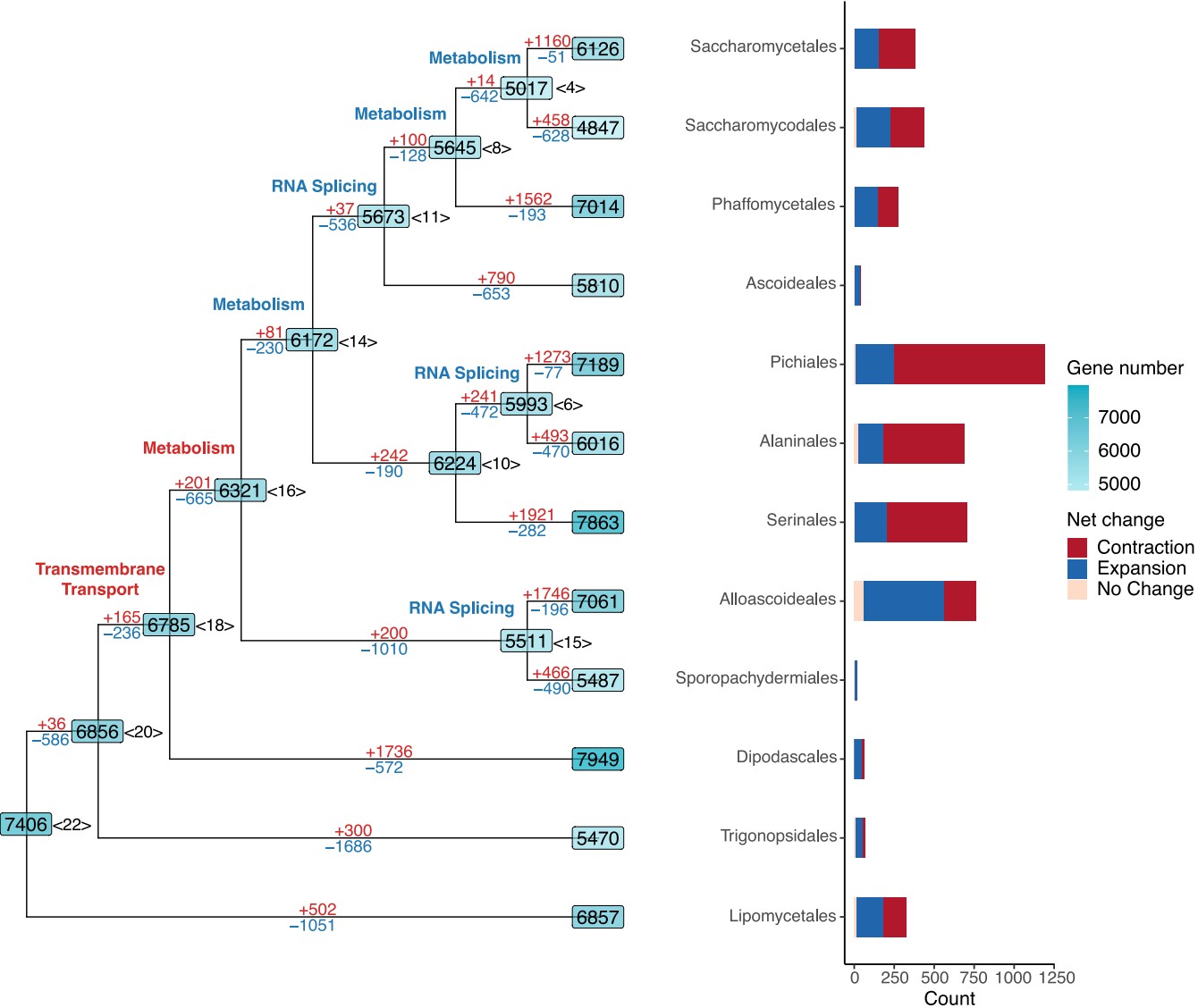

**Figure 5.  Yeasts have undergone a complex evolutionary history of gene families.**

The branches following the MRCA of each order have been collapsed to simplify the tree structure. Gene counts are marked on each node, with the corresponding node label positioned to its right. Gene gains are highlighted in red, while losses are depicted in blue along each branch. In addition, branches are annotated with key terms from enriched GO terms (P ≤ 0.05); here, red signifies gene family expansion, and blue denotes contraction. A bar plot to the right of the tree quantifies the net changes in gene families within the phylogeny after the MRCA of each order. The y-axis, labeled "count", reflects the number of gene families that underwent net changes—categorized into expansion, contraction, or no change. Expansion of a gene family is defined by a sum of net changes in copy number across all branches of an order being greater than 0, while contraction is defined by a sum less than 0, and no change is defined as a net change equal to 0.

Dipodascales FEL species have significantly altered their metabolic capacities.

## Some functional categories undergo waves of gains and losses

Ancestral reconstructions of gene family content revealed waves of gains and losses (Fig. 5), with a general trend of net gene loss from the Saccharomycotina common ancestor (SCA) to the most recent common ancestor (MRCA) of each order (tips in the Fig. 5, hereafter only use order names instead). The exception was Dipodascales, which experienced a net gain of 543 genes. Certain

nodes underwent notable changes in gene number; for instance, ancestral nodes such as <15>, Lipomycetales, and Trigonopsidales lost over 1000 genes each, whereas the Alloascoideales, Dipodascales, Phaffomycetales, Pichiales, Serinales, and Saccharomycetales ancestors gained over 1000 genes each.

Gene families within functional categories highlighted in previous analyses showed significant contractions and losses at ancestral yeast nodes. Specifically, gene families related to RNA splicing underwent substantial contractions at ancestral nodes <6>, <11>, <15>, Lipomycetales and Trigonopsidales, while expansions were observed at ancestral nodes Alaninales and Trigonopsidales (Fig. 5 and Dataset EV4). Gene families involved in metabolism

underwent frequent shifts, characterized by large expansions early in yeast evolution followed by subsequent contractions in some lineages. For instance, gene families related to amino acid, carbohydrate, vitamin, and sulfur metabolism showed significant expansions at ancestral nodes <20>, <18>, and <16> (Fig. 5 and Dataset EV4). In contrast, fatty acid, organic acid, and carbohydrate metabolism experienced notable contractions at ancestral nodes <4>, <8>, Ascoideales, Lipomycetales, Saccharomycodales, Serinales, Sporopachydermiales, and Trigonopsidales (Fig. 5 and Dataset EV4). Gene families associated with transcription also exhibit a complex evolutionary history, showing contractions at ancestral nodes <14>, Ascoideales, Lipomycetales, Serinales, Sporopachydermiales, and Trigonopsidales, and expansions at ancestral nodes <4>, <16>, <18>, Alaninales, Alloascoideales, and Lipomycetales (Dataset EV4).

To investigate the evolutionary trends of gene families that experienced significant contractions or expansions in CAFE analyses within each yeast order, we calculated the net change of these gene families (net gain or loss across all branches). In orders that include Alaninales (508/689), Pichiales (943/1194) and Serinales (498/704), over 70% of gene families with net changes experienced contractions, while in Alloascoideales (507/762), 66% of the events were gene family expansions (Fig. 5). The remaining orders exhibited a nearly balanced mix of gene family expansion and contraction events. Gene families with net expansions were enriched in plasma membrane and transmembrane transporter-related GO terms (Dataset EV5). Conversely, DNA polymerase activity was prevalent in some gene families undergoing contractions, except in Serinales and Trigonopsidales, which are enriched in ligase activity and DNA repair functions, respectively (Dataset EV5).

To explore novel genes gained in the most recent common ancestor of each order, we selected orphan gene families (i.e., order-specific gene families) as determined by the coverage of each gene family across each order. Examination of orphan genes revealed variation among orders. Alloascoideales and Sporopachydermiales orders each possessed over 180 orphan gene families, while other orders had fewer than 80 (Appendix Fig. S6). The Dipodascales and Trigonopsidales orders each had only two orphan gene families, while Pichiales had one. Orphan genes were not enriched in specific functional categories. After identifying potential homologous proteins of orphan genes in the NCBI non-redundant (NR) protein database, we found that 24,577 orphan genes (96.5%) appear to have emerged de novo. Only 36 genes (0.2%) likely originated from speciation or duplication events following the most recent common ancestor of Saccharomycotina, while 865 genes (3.4%) are likely the result of horizontal gene transfer (Appendix Fig. S7).

# Discussion

Examination of gene family evolution of 1154 genomes of nearly all known Saccharomycotina species elucidated, for the first time ever, the landscape of gene family evolution across a eukaryotic subphylum. Reductive evolution emerges as the main theme, marked by a transformation from a versatile SCA to descendants with more specialized lifestyle/metabolic capacity (Shen et al, 2018) and smaller gene repertoires (Fig. 5). In extant species, most yeasts exhibited similar weighted average gene family sizes and

evolutionary rates. However, significant differences were observed in FELs compared to their SEL relatives in several independent yeast orders. The gene family size differences between FELs and SELs, enriched in similar functional categories, suggest that the same evolutionary trajectory has occurred repeatedly and independently in multiple yeast orders, indicating a broader trend rather than isolated incidents. The FELs demonstrated notable contractions and losses in gene families, especially those related to RNA splicing and the DASH complex (Fig. 3B,D). Alterations in the pre-mRNA splicing pathway could generate novel transcript variants, potentially allowing some yeasts to better respond to environmental changes (Wahl et al, 2009; Liu et al, 2022). In addition, impairments in the DASH complex may cause genomic instability, which, although potentially harmful under stable conditions, might provide adaptive advantages in fluctuating environmental stresses by increasing genetic diversity (Jenni and Harrison, 2018; Boyko and Kovalchuk, 2011).

These gene family contractions and losses in FELs may contribute to their higher evolutionary and speciation rates (Figs. 2C and 3E) by enabling rapid genomic adaptations that optimize cellular processes crucial for survival and reproduction in diverse and challenging environments. For example, the FEL of Dipodascales is primarily found in the Arthropoda environment (Opulente et al, 2024), which is partially characterized by the production of various antifungal compounds and generally hostile conditions for many microorganisms (Kett et al, 2021; Stefanini, 2018). This lineage also shows significant contractions in gene families related to metabolism and a general loss of growth traits, with a notable exception being the acquisition of glycerol utilization abilities (Fig. 3B,F). This capability could be a key adaptation allowing them to thrive in specialized environments. Interestingly, a similar adaptation has been observed in endosymbionts like *Buchnera aphidicola* in aphids and *Wigglesworthia glossinidia* in flies, both of which effectively utilize glycerol (Zientz et al, 2004). The expansion of cytochrome P450 and cytochrome c oxidase assembly protein subunit gene families in Saccharomycodales and Dipodascales FELs (Dataset EV3) suggests enhanced detoxification and metabolism of xenobiotic compounds, supporting their adaptation to hostile environments (Esteves et al, 2021; Durairaj et al, 2016; Kagan et al, 2009). CAFE analysis has shown that certain functional categories, such as RNA splicing, metabolism, and cytochrome P450, are affected at more ancestral nodes in the yeast phylogeny (Fig. 5 and Dataset EV4). This suggests that the similar evolutionary trajectory observed across multiple yeast orders may be influenced by reductive evolution throughout the evolutionary history of yeast.

FELs exhibited a greater number of gene losses compared to SELs; however, these losses were not frequently associated with specific trait losses. For example, while six genes in the pre-mRNA splicing pathway were lost in Saccharomycodales, no notable gene loss was observed in Trigonopsidales (Appendix Table S3). Interestingly, Trigonopsidales displayed more intron loss than Saccharomycodales (Fig. EV5). These findings raise fundamental questions in evolutionary biology: Is gene loss adaptive, neutral, or deleterious? Does gene loss precede or follow trait changes? In some cases, gene loss may confer advantages by reducing metabolic costs (Pande et al, 2014; D'Souza and Kost, 2016) or enhancing microbial pathogenicity (Albalat and Cañestro, 2016; Domergue et al, 2005). Conversely, it can impair essential functions, reducing

fitness (Martí-Solans et al, 2016). Functional redundancy often mitigates the impact of gene loss, highlighting the role of evolutionary pressures in shaping gene content and functionality (Zhang, 2012; Hanada et al, 2009; Mendonça et al, 2011). The order of gene loss and trait change can vary: in some cases, gene loss initiates phenotypic shifts that drive innovation or adaptation (Sánchez-Serna et al, 2024; Martí-Solans et al, 2021), while in others, trait changes render certain genes redundant, leading to their eventual loss (Lynch and Conery, 2000; Stern, 2013). Whether intron loss drove the reduction of splicing machinery (or the other way round) remains an open question. On the one hand, the "use it or lose it" principle suggests that as introns become fewer, maintaining the full complement of splicing genes may be metabolically costly or functionally unnecessary, leading to their gradual loss (Cech and Steitz, 2014; Lynch and Conery, 2003). Conversely, the partial or complete loss of splicing factors can undermine intron processing, making introns dispensable and accelerating their decay (Krämer, 1996; Nilsen and Graveley, 2010). Resolving cause and effect requires robust comparative genomic data alongside functional assays: mapping the phylogenetic distribution of introns and associated splicing components can reveal the sequence of events, while experimental studies can illuminate how the loss of splicing factors impacts intron stability. Ultimately, both scenarios may be valid under different ecological and evolutionary contexts, reflecting the interplay of selection pressures, genome streamlining, and the redundancy of splicing pathways.

When comparing gene family sizes among extant and ancestral yeasts, gene family contraction events are consistently observed. This is evident in comparisons of FELs vs. SELs (Fig. 3) and in several ancestral branches near the root of the yeast phylogeny (Fig. 5). In contrast, other fungal phyla, like other eukaryotes, underwent significant gene duplications, such as whole-genome duplications (WGD) and tandem duplications, which were essential in shaping gene repertoires (Lynch and Conery, 2000; Marcet-Houben and Gabaldón, 2015; Albertin and Marullo, 2012; Van de Peer et al, 2009). These duplications provided the genetic foundation for the diversification of key metabolic, signaling, and developmental pathways (Wisecaver et al, 2014; Blanc and Wolfe, 2004; Panchy et al, 2016). Other fungal phyla, such as chytrids and zygomycetes, expanded gene families related to nutrient acquisition, cell wall biosynthesis, and environmental sensing (Merényi et al, 2023; Sun et al, 2011; Chang et al, 2022; Corrochano et al, 2016). Other eukaryotic groups also experienced large-scale duplications, with plants evolving processes like photosynthesis and stress responses, and animals diversifying immune, sensory, and multicellular systems (Ocaña-Pallarès et al, 2022; One Thousand Plant Transcriptomes Initiative, 2019; Blanc and Wolfe, 2004; Zou et al, 2009; Buckley and Rast, 2015; Sánchez-Gracia et al, 2009). In contrast, the widespread gene loss observed in many ancestral yeast branches (Fig. 5) (Shen et al, 2018), coupled with shift from metabolic generalism to specialism (Opulente et al, 2024), likely reflects the continuation of early fungal evolutionary patterns. This trajectory of loss in yeasts contrasts with the evolutionary trajectories seen in other fungal and other eukaryotic lineages.

Fungi also exhibit genomic diversity shaped by both gradual and episodic gene family contractions and duplications (Merényi et al, 2023). Gradual changes typically result from sustained ecological

pressures, fine-tuning metabolic and regulatory networks to adapt to specific environments or host interactions (Tedersoo et al, 2014; Naranjo-Ortiz and Gabaldón, 2019; Aylward et al, 2017; Verbruggen and Toby Kiers, 2010). Episodic bursts, often associated with major events like WGDs or horizontal gene transfer (HGT), lead to rapid diversification and the acquisition of new functions (Corrochano et al, 2016; Albertin and Marullo, 2012; Richards et al, 2011). These bursts are frequently linked to ecological disruptions or shifts in environmental conditions. Together, gradual adaptation and episodic genetic events contribute to the genomic complexity of fungi, reflecting a dynamic balance between long-term adaptation and rapid evolutionary innovation.

State-of-the-art evolutionary genomic and phylogenomic studies now routinely report or analyze genomic data from hundreds to thousands of genomes (Christmas et al, 2023; Opulente et al, 2024; One Thousand Plant Transcriptomes Initiative, 2019; Shen et al, 2020), ushering us in the "Thousand Genomes Era". Analyzing gene families across thousands of genomes presents substantial challenges, including handling large datasets, accurately identifying and comparing complex genomic variations, and offering detailed functional annotations for a diverse range of genes. Traditional gene family analyses often concentrate on specific gene families, species, and gene family size evolution, leading to a gap in large-scale comparative analysis. In this study, we developed a comprehensive approach to explore gene family size differences across and within yeast lineages and compare gene family evolution between yeasts and three other ancient lineages (plants, animals, and filamentous fungi). By calculating weighted average gene family size, comparing evolutionary rates, and performing statistical tests, we characterized gene family dynamics, such as expansions, contractions, and losses. These analyses provided insights into the evolutionary pressures shaping gene family composition and allowed for the reconstruction of ancestral gene family histories across over 1000 genomes. This approach establishes a comparative framework that can be applied to other major branches of the tree of life.

# Methods

**Reagents and tools table**

| Reagent/Resource | Reference or Source | Identifier or Catalog Number |
|---|---|---|
| **Experimental models** | | |
| N/A | | |
| **Recombinant DNA** | | |
| N/A | | |
| **Antibodies** | | |
| N/A | | |
| **Oligonucleotides and other sequence-based reagents** | | |
| N/A | | |
| **Chemicals, enzymes, and other reagents** | | |
| N/A | | |
| **Software** | | |
| CD-HIT 4.8.1 | https://github.com/weizhongli/cdhit | |

| Reagent/ Resource | Reference or Source | Identifier or Catalog Number |
|---|---|---|
| OrthoFinder 3.0 | https://github.com/davidemms/ OrthoFinder | |
| InterProScan 5.59 | https://github.com/ebi-pf-team/ interproscan | |
| eggNOG-mapper 2.1.9 | https://github.com/eggnogdb/eggnog-mapper | |
| GhostKOALA 2.0 | https://www.kegg.jp/ghostkoala/ | |
| KofamKOALA | https://www.genome.jp/tools/ kofamkoala/ | |
| R 4.3 | https://www.r-project.org/ | |
| IQ-TREE 2.2.3 | https://github.com/iqtree/iqtree2 | |
| diptest 0.76.0 | https://cran.r-project.org/web/ packages/diptest/index.html | |
| dbscan 1.1.12 | https://cran.r-project.org/web/ packages/dbscan/index.html | |
| ape 5.8-1 | https://cran.r-project.org/web/ packages/ape/index.html | |
| BayesTraits 4.0.0 | https://isu-molphyl.github.io/EEOB563/ computer_labs/lab8/BayesTraits.html | |
| BLAST 2.15.0+ | https://github.com/ncbi/ blast_plus_docs | |
| InParanoid 4.2 | https://bitbucket.org/ sonnhammergroup/inparanoid/src/ master/ | |
| CAFE 5.0 | https://github.com/hahnlab/CAFE5 | |
| clusterProfiler 4.6.0 | https://github.com/YuLab-SMU/ clusterProfiler | |
| GOATOOLS 1.2.3 | https://github.com/tanghaibao/goatools | |
| ggtree 3.8.0 | https://github.com/YuLab-SMU/ggtree | |
| ggplot2 3.4.3 | https://github.com/tidyverse/ggplot2 | |
| cowplot 1.1.3 | https://github.com/wilkelab/cowplot | |
| Adobe Illustrator CC 2024 | https://www.adobe.com/ | |
| Other | | |
| N/A | | |

## Data collection and collation

For our study on gene family evolution within Saccharomycotina yeasts, we acquired a comprehensive dataset comprising 1154 Saccharomycotina yeast genomes. In addition, 21 non-budding yeast species were sampled as outgroups based on current understanding of Ascomycota phylogeny. These genomes, along with their annotations and a species tree, were obtained from our previous study (Opulente et al, 2024). This dataset provides a robust foundation for examining the evolutionary dynamics of gene families in yeasts. To compare the tempo and mode of gene family evolution of yeasts to other major eukaryotic lineages, we expanded

our dataset to include 761 filamentous ascomycetes (Pezizomycotina) genomes (Shen et al, 2020), 1178 plant genomes and transcriptomes (One Thousand Plant Transcriptomes Initiative, 2019), and 83 animal genomes (Liu et al, 2024), including gene annotations for each. For all genomes, we kept the amino acid sequence translated from the longest protein coding sequence (CDS) from each gene. For plant transcriptomes, we adopted a protocol from (One Thousand Plant Transcriptomes Initiative, 2019), using cd-hit version 4.8.1 (Li et al, 2001) with a 99% sequence identity threshold to minimize redundancy. NCBI taxonomy and source information of all genomes and transcriptomes included in this study are also provided in Dataset EV1 and the Figshare repository. Saccharomycotina species names in the supplementary tables and the Figshare repository were the current species names at the time used in the recent study (Opulente et al, 2024). For synonymous names and recent taxonomic updates, we refer the reader to the online MycoBank database.

## Delineation of gene family and functional annotation

To infer a comprehensive profile of gene families in budding yeasts, we delineated groups of orthologous genes (orthogroups, hereafter referred to as gene families) for the Saccharomycotina yeast dataset using OrthoFinder version 3.0, with default settings (Emms and Kelly, 2015). Following the approach of previous studies (Cheng et al, 2023; Ma et al, 2021; Trouern-Trend et al, 2020), we used orthogroups from OrthoFinder as gene families. For consistency, we applied the same method to categorize gene families in Pezizomycotina, animal, and plant datasets. Due to the large number of genomes and transcriptomes in plants, we initially processed protein sequences from 31 representative genomes with the "-core" parameter to establish base orthogroups, and subsequently classified the protein sequences from an additional 1148 transcriptomes using the "-assign" parameter.

To obtain functional information of yeast gene families, we annotated all yeast genes from three independent aspects, including InterPro protein domains, and Gene Ontology (GO) and Kyoto Encyclopedia of Genes and Genomes (KEGG) terms. InterPro annotations were generated using InterProScan as part of a previous study (Opulente et al, 2024). GO annotations were generated using the eggNOG-mapper version 2.1.9 (Cantalapiedra et al, 2021) with the search mode set to "mmseqs". We initially compared KEGG annotations using the web-based GhostKOALA version 2.0 (Kanehisa et al, 2016) with the KofamKOALA based annotations used in the study (Opulente et al, 2024). Due to GhostKOALA providing annotations for a larger number of gene families, we ultimately chose to exclusively use GhostKOALA for our final KEGG annotations.

## Weighted average gene family size analysis of gene family evolution

To assess the variations in gene family size among yeasts, Pezizomycotina, animals, and plants, we calculated the weighted average gene family size using a custom R script according to the following formula described in (One Thousand Plant Transcriptomes Initiative, 2019). The weighted average gene family size represents the overall gene family size of a species within a group,

accounting for the relative sizes of individual gene families.

$$\text{Weighted Avg.Size} = \frac{\sum_{i=1}^{n} \text{copy}_i \times w_i}{n} \times \text{mean max} \qquad (1)$$

Taking yeasts as an example, in the formula, $n$ represents the total number of gene families in the dataset. For each gene family $i$, we calculate its maximum copy number across all 12 orders of the Saccharomycotina simultaneously, denoted as $\max(copy_i)$. We then use the inverse of this maximum copy number as a weight for that gene family $w_i = \frac{1}{\max(copy_i)}$. *mean max* is the average of the maximum copy numbers of these $n$ gene families. We also explored alternative weighting methods, including: (1) excluding the top 5% of values and using the next highest value as the weight, (2) using the mean value as the weight, and (3) using the median value as the weight. The primary analyses were repeated with these methods, and the overall patterns in yeast weighted average gene family size comparisons remained consistent with the original results (Appendix Figs. S8 and S9).

Our preliminary analysis revealed a large number of gene families with highly restricted taxon distribution, which may confound the calculation of weighted average gene family size. Therefore, we implemented a lineage-based coverage assessment method (Merényi et al, 2023) for gene families across different taxa to exclude species-specific gene families. Specifically, we focused on assessing the coverage of each gene family within these 4 distinct groups, using yeasts as an example. Coverage in this context refers to the proportion of species within each clade that possesses a particular gene family. Using yeasts as an example, for each gene family, we first calculated its coverage in each of the 12 yeast orders (Groenewald et al, 2023), and then took the average value as the overall coverage of the gene family. Similar procedures were followed for Pezizomycotina (9 classes), animals (14 phyla), and plants (22 phyla). Gene families with low average coverage values are likely to be highly species-specific. Given a bimodal distribution in the density plots of average coverage for gene families, we established a relaxed threshold of 0.1 to identify species-specific gene families. Families with average coverage below this threshold were considered species-specific for further analysis. This exclusion criterion was applied uniformly across the 4 groups studied.

To robustly test the correlation between weighted average gene family sizes and gene counts while accounting for phylogenetic relationships, we first converted the data into phylogenetic independent contrasts (PICs) using the "pic" function from the R package ape version 5.7.1, based on the respective phylogenetic trees for yeasts, filamentous ascomycetes, animals, and plants. Phylogenetic trees were obtained from previous studies (Liu et al, 2024; Shen et al, 2020; Opulente et al, 2024) and pruned to include only the species we studied using gotree version 0.4.4 (Lemoine and Gascuel, 2021). In the previous study (One Thousand Plant Transcriptomes Initiative, 2019), the plant phylogenetic tree was constructed using ASTRAL, which is not optimized for accurate branch length estimation. Therefore, we retained the original tree topology and protein sequences from the previous study to reconstruct branch lengths using IQ-TREE version 2.2.3 (Kalyaanamoorthy et al, 2017). We then conducted a Spearman correlation test between these transformed datasets using the cor.test function (method = spearman) from the R package stats version 4.3.2. This method was also applied to examine the correlation between PICs of weighted average gene family sizes and genome sizes.

## Classification of faster-evolving and slower-evolving lineages

To examine the variation in the weighted average gene family size within 12 orders, we utilized the R package diptest version 0.76.0 for conducting unimodality tests separately on the evolutionary rates (measured as the branch length from the tip to the Saccharomycotina common ancestor (SCA) on the phylogenetic tree) and the weighted average gene family sizes for each of the 12 orders. In addition, we applied the same method to analyze the branch length from the tip to the most recent common ancestor of the order in focus, which yielded the same results. For orders exhibiting significant non-unimodal distributions in both evolutionary rates and weighted average gene family sizes, we applied density-based spatial clustering of applications with noise (DBSCAN) algorithm using the R package dbscan version 1.1.12 to identify clusters based on evolutionary rates. In addition, we mapped weighted average gene family sizes onto the phylogenetic tree to examine lineage-specific variations. In orders displaying lineage-specific variations, the DBSCAN clusters with faster evolutionary rates were labeled as faster-evolving lineages (FELs), and those with slower rates were identified as slower-evolving lineages (SELs).

## Analysis of gene family expansion and contraction between faster and slower evolving lineages

To determine which gene families exhibited expansion or contraction in FELs compared to their SEL relatives, we performed a fold change analysis using a custom R script, based on the method developed in the previous study (One Thousand Plant Transcriptomes Initiative, 2019). For a given yeast order with FEL and SEL lineages, we first calculated the average copy numbers (non-weighted) for each gene family within the FELs and SELs, respectively, then divided the average value of FELs by that of SELs. In addition, we performed the Kolmogorov–Smirnov (KS) test using the ks.test function from the R package stats version 4.3.2, coupled with the Bonferroni method for $p$-value adjustment, to ascertain the significance of these expansions or contractions. Consistent with the criteria established in prior research (One Thousand Plant Transcriptomes Initiative, 2019), we reported those gene families that underwent significant changes (adjusted $P \leq 0.05$), and a fold change exceeding 1.5 for expansions or less than 0.67 for contractions. A fold change of 0 was interpreted as a loss of the gene family, while a fold change nearing positive infinity indicated the acquisition of a gene family.

To verify that the gene families identified as significantly changed (expansions/gains or contractions/losses) in the fold change analysis exhibited similar evolutionary trends relative to ancestral yeasts, we examined their ancestral gene family copy numbers using CAFE results. Specifically, we compared the average copy numbers of FELs to those of the MRCA for each order. Across all three orders, most gene families exhibited similar evolutionary patterns (Appendix Fig. S10).

## Principal component analysis of gene family presence and absence pattern

To compare the difference of gene family composition across yeasts, we conducted a Principal Component Analysis (PCA) based on the presence (1) or absence (0) data of gene families (Merényi

et al, 2023). We first discerned conserved and species-specific gene families by setting average coverage threshold at 0.5 based on the density plot (Fig. EV1). Gene families with the average coverage equal to or exceeding 0.5 were considered conserved, while those below the threshold were classified as species-specific. We employed PCA on both conserved and species-specific gene families using the R package stats version 4.3.2. Consequently, we performed density clustering to the PCA results using the dbscan function from the R package dbscan version 1.1.12, grouping species with similar distribution patterns into distinct clusters. We also conducted the same analysis using a more relaxed threshold of 0.1 in the PCA to exclude more noise from species-specific and/or rare gene families, which yielded consistent results.

To identify key gene families driving the distribution of yeasts along the first or second principal components, we employed a custom R script for detailed analysis (Appendix Fig. S11). Initially, we ranked gene families according to their contribution (from the rotation table in the PCA results using the R package stats) to each principal component (PC), both in ascending and descending order. To identify the optimal number of top ranking gene families whose average presence values best correlate the coordinates of yeasts, we calculated the average presence values for the top 1, 2, i, and up to top $n$ gene families (where i is the specific number of gene families, and $n$ is the total number of gene families). The average presence value for the top $n$ gene families was determined by dividing the total presence of these $n$ gene families in a species by $n$. Subsequently, we conducted a Spearman correlation test using the cor.test function (method = spearman) from the R package stats version 4.3.2. This test assessed the relationship between the average presence values of species in the top i gene families and their respective positions on the PC. The gene families with the highest absolute correlation values were selected. Since we calculated the correlation between PC coordinates and the average copy numbers of the top i gene families, a positive correlation indicates that species with higher PC coordinates tend to possess more copies of the top i gene families, while a negative correlation suggests that species with higher PC coordinates are likely to have fewer copies of these gene families.

To assess the robustness of our methods for detecting yeast genomes/lineages with significant changes in gene content and identifying the major gene families responsible, we performed PCA on the copy number data matrix and PCoA on both the presence/absence and copy number data. Both PCA and PCoA performed better on the presence/absence data, and they showed similar patterns in identifying key gene families that explain the distribution of yeasts on PCA/PCoA scatter plots (Appendix Figs. S12–S16). Although PCA has theoretical limitations when applied to binary (presence/absence) data, our findings are validated by further evidence (Fig. 4) and align with results from alternative approaches.

### Assessing the impact of HGT genes

In our previous research on yeast genome evolution, we found that yeasts exhibit very low levels of HGT, with approximately 0.04% to 0.06% of genes putatively acquired via HGT (Shen et al, 2018). To ensure that HGT does not influence the results of our current study, we excluded HGT-related gene families by using BLASTP to filter out the 878 HGT-acquired genes identified in our previous work (Shen et al, 2018). We then reran the key analyses, including fold-change and PCA, to evaluate the impact of these exclusions on our findings. The results confirmed that the exclusion of HGT-related genes did not affect the overall conclusions of the study (Appendix Figs. S17–S19).

### Analysis of the rates of speciation and carbon source utilization trait gain and loss

To investigate whether different carbon source utilization traits are more readily acquired or lost in the FELs or SELs, we used the analytical method and carbon source utilization data from previous studies (Opulente et al, 2024). Firstly, we pruned the species tree to only retain yeasts with available metabolic data, resulting in trees comprising exclusively Dipodascales FEL or SEL species. Subsequently, we employed BayesTraits version 4.0.0 and its reverse jump model (Pagel and Meade, 2006) to conduct two simulations for each carbon source. The first simulation set the loss rate of carbon source utilization traits equal to the acquisition rate (using the parameter "Res q01 q10"), while the second did not equate these rates (no specific parameter used). In addition, each model underwent 10,100,000 iterations, using 200 stepping stones, with sampling every 1000 iterations. The burn-in was set at 100,000 iterations. We also employed the R package coda version 0.19.4 for visualization purposes to ensure model convergence.

To select the appropriate model for determining whether the loss rate of carbon source utilization traits should be equal to or different from the acquisition rate, we calculated Log Bayes Factors according to the BayesTraits manual (https://www.evolution.reading.ac.uk/BayesTraitsV4.1.1/BayesTraitsV4.1.1.html). Log Bayes Factors is utilized to compare the relative evidence between two statistical models. When the Log Bayes Factor is less than 2, we opt for the relatively simpler model (where the acquisition rate is equal to the loss rate). Conversely, when it is 2 or higher, we select the more complex model (where the acquisition rate is not equal to the loss rate). Subsequently, based on the selected model, we count the number of instances where the carbon source utilization trait's acquisition rate is either greater than or less than its loss rate. If the instances of the acquisition rate being higher than the loss rate significantly outnumber those where it is lower, we conclude that the lineage tends to acquire that particular carbon source utilization trait. On the other hand, if there are more instances of the acquisition rate being lower than the loss rate, the lineage is considered more inclined to lose that trait. If the simpler model is chosen based on the Log Bayes Factor, we infer that the lineage is neither inclined to lose nor to acquire the carbon source utilization trait.

To investigate the connections among gene family expansions and contractions, the acquisition and loss of carbon traits, and the diversification of species, we estimated speciation rates from the DR statistic (Title and Rabosky, 2019; Jetz et al, 2012) calculated using the inverse equal splits method (Redding and Mooers, 2006) using a recently published time-calibrated phylogeny (Opulente et al, 2024).

### Investigation of metabolic pathways, the spliceosome pathway, and the DASH complex

To investigate how gene loss might affect crucial biological processes in the FEL of Dipodascales, we used *S. cerevisiae* as a reference to map gene names to its pre-mRNA splicing pathway, metabolic pathways, and the DASH complex. We first identified

gene families that exhibited significant contraction or loss in the fold change analysis and those that were representative in contributing to the principal component, using the representative genes from *S. cerevisiae*. If an *S. cerevisiae* gene was assigned to a gene family according to OrthoFinder results, we named the gene family using the *S. cerevisiae* gene name. Otherwise, the gene family remained unnamed due to the uncertainty of its classification. Subsequently, we used these gene family names for pathway mapping. Specifically, we used the search function on the KEGG website (https://www.genome.jp/pathway/sce03040) for the pre-mRNA splicing pathway, the Highlight Gene (s) feature on the *Saccharomyces* Genome Database (SGD) (Wong et al, 2023) biochemical pathways site (https://pathway.yeastgenome.org/overviewsWeb/celOv.shtml) for metabolic pathways, and the previous study (Jenni and Harrison, 2018) for DASH complex.

To verify the absence of genes indicated in our gene copy numbers heatmap (Fig. 4A), we carried out independent orthology delineation using InParanoid version 4.2 (Remm et al, 2001) and sequence search using Basic Local Alignment Search Tool (BLAST) version 2.15.0+ (Altschul et al, 1990). This was to ensure accuracy and address potential misassignments by OrthoFinder or errors in genome annotations. With InParanoid, we compared the protein-coding genes from all species in our heatmap against those of *S. cerevisiae* to identify orthologous genes, confirming that species depicted as lacking certain genes genuinely did not have those orthologs. We also used blastp (e-value threshold of 1e−5) to compare species, which are shown as missing gene families in the heatmap, with a reference species that contained all genes in our heatmap. For addressing potential annotation inaccuracies, we performed genome-protein comparisons using tblastn (also with an e-value of 1e−5).

## CAFE analysis of gene copy number evolution

To estimate gene family expansion and contraction events, we utilized computational analysis of gene family evolution using (CAFE) version 5.0 (Mendes et al, 2021). Due to the computational limitation of CAFE in processing the complete analysis of 1154 genomes, we first employed separate analyses for each of the 12 orders. For these analyses, the input time tree was pruned to include only species from the order under study. The input gene families needed to meet any of the three criteria: (1) presence in the MRCA of studied order, as determined by maximum parsimony; (2) presence in the studied order and at least one of the remaining 11 orders; and (3) presence in both the studied order and the outgroup. Gene families not meeting those three criteria are specific to the order under study, and thus are irrelevant for the CAFE analysis of 12 order MRCAs. The estimated gene contents of each yeast order was then analyzed by CAFE to reconstruct gene family copy numbers at the SCA. The input time tree was pruned to only include the MRCAs of each of the 12 yeast orders.

We experimented with different numbers of gamma categories ($k \in [2, 10]$) using the "-k" parameter and selected the k value with the highest likelihood. To determine the alpha (the evolutionary rate of genes within gene families over time) and lambda (the rate of increase or decrease of gene families over time) values, we ran 10 iterations with the determined k value and chose the alpha and lambda values that yielded the maximum likelihood.

To confirm the reliability of our CAFE analysis on the full dataset of 1154 species, we used the same methods on subsampled datasets of 200 species and 50 species. The 200 and 50 species datasets were subsampled based on genome completeness from BUSCO results, ensuring that at least one species from each order was included.

To ensure robust reconstruction of ancestral node gene contents, we only displayed gene families that met any of the following criteria: (1) present in the SCA, as determined by maximum parsimony; and (2) present in only a specific order and the MRCA of that order.

## Orphan gene families

We defined orphan gene families as those specific to a particular order and exhibiting a high species coverage within that order. Specifically, an orphan gene family is characterized by being present in at least 98% (Groenewald et al, 2023) of the species within a given order. This means that to qualify as an orphan, a gene family must be found in 98% or more of the species within the order under consideration. In addition, these gene families must be completely absent in all other remaining orders.

To determine the origin of orphan genes, we performed two BLASTP comparisons. First, we compared orphan genes against our dataset of 1154 yeast genomes using thresholds established in a prior study on sequence homology searching (Pearson, 2013): e-value < $10^{-5}$, bit score >50, and percent identity >40%. Second, we compared orphan genes against the NCBI non-redundant (NR) protein database using thresholds of e-value < $10^{-5}$ and bit score >50. If a homologous protein was identified in the first comparison outside the respective order, the orphan gene was inferred to have originated from duplication or speciation followed by rapid sequence divergence. If no homolog was found in the first comparison but was detected in the second, the gene was inferred to have potentially originated via horizontal gene transfer. Orphan genes without homologs in either comparison were classified as putatively de novo.

## Functional enrichment analysis

We conducted functional enrichment analyses of gene families across fold change, PCA, and CAFE analyses. For the fold change analysis, the background set for enrichment consisted of the union of gene families present in all yeast species within the studied order. In PCA, particularly for the top 610 gene families linked with PC1, the background was composed of all gene families involved in the PCA. For the CAFE analysis, the background for enrichment was the set of gene families included in the gene family copy number table used as input. Our enrichment analyses drew upon various annotations, including GO annotations, KEGG annotations, and InterPro annotations. The correspondence description tables for GO terms and KOs and InterPro entries were downloaded from the GO (https://geneontology.org/), KEGG (https://www.genome.jp/kegg/) and InterPro (https://www.ebi.ac.uk/interpro/) websites, respectively, on November 23, 2023.

All enrichment analyses were conducted using the R package clusterProfiler version 4.6.0 (Wu et al, 2021) with default parameters, selecting only results with $P \leq 0.05$. To translate GO terms into more generalized and concise GO slims in fold change enrichment analysis, we employed GOATOOLS version 1.2.3 (Klopfenstein et al, 2018). For this process, we utilized the go-

basic.obo and goslim_yeast.obo files, which were retrieved from the Gene Ontology website on December 13, 2023.

## Data visualization

We utilized the R package ggtree version 3.8.0 (Xu et al, 2022) to visualize phylogenetic trees and associated CAFE data, and ggplot2 version 3.4.3 for other graphs. Images representing taxa were hand-drawn, sourced from PhyloPic (https://www.phylopic.org/), and customized in terms of color using rphylopic version 1.3.0.

## Data availability

The reference phylogeny of yeasts, along with genome and annotation data for yeasts, Pezizomycotina, animals, and plants, are accessible from previous studies described above. In addition, NCBI taxonomy and source details for this study can be found in Dataset EV1. We have deposited all new functional annotations, analyses, and codes in the Figshare repository at https://figshare.com/s/66d97c17e16c241f41e6. You can also find the code on Github at https://github.com/vnuii/yeast-gene-family.

The source data of this paper are collected in the following database record: biostudies:S-SCDT-10_1038-S44320-025-00118-0.

## Peer review information

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

## Acknowledgements

This study was supported by the National Key R&D Program of China (2023YFA0915500), National Natural Science Foundation of China (42376147), Intramural Joint Program Fund of State Key Laboratory of Microbial Technology (SKLMTIJP-2024-03), Ocean Negative Carbon Emissions (ONCE) Program and Shandong Provincial Natural Science Foundation for Distinguished Young Scholars (ZR2024JQ027). XZ was supported by grants from the Basic and Applied Basic Research Foundation of Guangdong Province (2022A1515010223) and the National Natural Science Foundation of China (32260652). JLS is a Howard Hughes Medical Institute Awardee of the Life Sciences Research Foundation. Research in the Hittinger Lab is funded by the National Science Foundation (DEB-2110403), USDA National Institute of Food and Agriculture (Hatch Project 7005101), in part by the DOE Great Lakes Bioenergy Research Center (DOE BER Office of Science DE-SC0018409), and an H. I. Romnes Faculty Fellowship (Office of the Vice Chancellor for Research and Graduate Education with funding from the Wisconsin Alumni Research Foundation). Research in the Rokas lab is supported by the National Science Foundation (DEB-2110404), NIH/National Institute of Allergy and Infectious Diseases (R01 AI153356), and the Burroughs Wellcome Fund.

## Author contributions

**Bo Feng**: Formal analysis; Investigation; Visualization; Writing—original draft. **Yonglin Li**: Investigation; Methodology. **Biyang Xu**: Investigation; Methodology. **Hongyue Liu**: Investigation; Methodology. **Jacob L Steenwyk**: Investigation; Methodology. **Kyle T David**: Investigation; Methodology. **Xiaolin Tian**: Investigation; Methodology. **Carla Gonçalves**: Investigation; Methodology. **Dana A Opulente**: Resources; Data curation. **Abigail L LaBella**: Resources; Data curation. **Marie-Claire Harrison**: Resources; Data curation. **John F Wolters**: Resources; Data curation. **Shengyuan Shao**: Resources; Data curation. **Zhaohao Chen**: Resources; Data curation. **Kaitlin J Fisher**: Resources; Data curation. **Marizeth Groenewald**: Resources; Data curation. **Chris Todd Hittinger**: Conceptualization; Resources; Supervision; Funding acquisition; Project administration. **Xing-Xing Shen**: Conceptualization; Resources; Supervision; Funding acquisition; Project administration. **Shengying Li**: Resources; Project administration; Writing—review and editing. **Antonis Rokas**: Conceptualization; Resources; Supervision; Funding acquisition; Project administration; Writing—review and editing. **Xiaofan Zhou**: Conceptualization; Resources; Supervision; Funding acquisition; Investigation; Writing—original

draft; Project administration. **Yuanning Li**: Conceptualization; Resources; Supervision; Funding acquisition; Investigation; Writing—original draft; Project administration.

Source data underlying figure panels in this paper may have individual authorship assigned. Where available, figure panel/source data authorship is listed in the following database record: biostudies:S-SCDT-10_1038-S44320-025-00118-0.

## Disclosure and competing interests statement

JLS is an adviser for ForensisGroup, Inc. AR is a scientific consultant for LifeMine Therapeutics, Inc. The other authors declare no other competing interests.

# Expanded View Figures

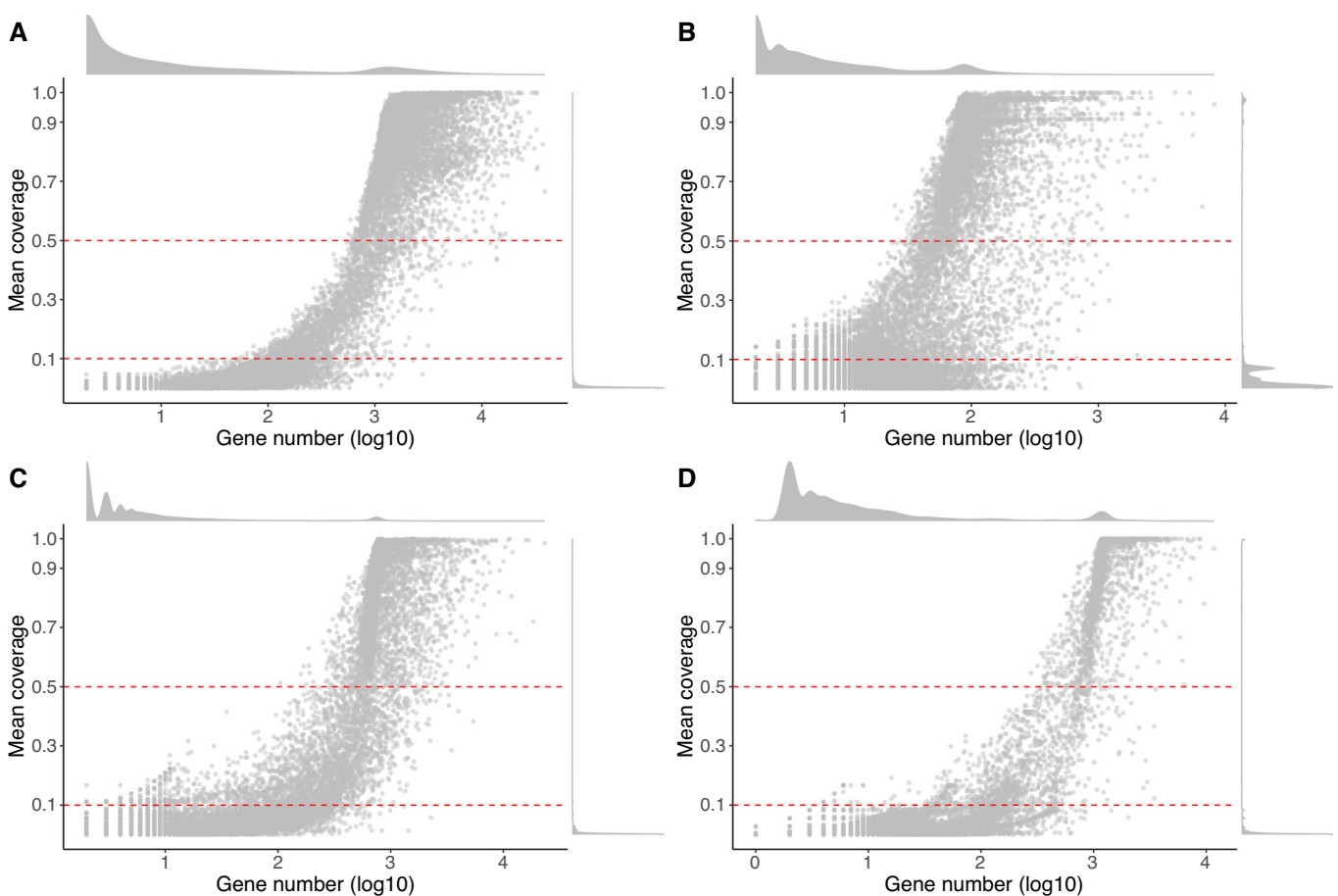

**Figure EV1.  Mean coverage of each gene family across 4 groups.**

Mean coverage represents the average coverage of gene families across clades: plants ($n = 21$), animals ($n = 14$), Pezizomycotina ($n = 9$), and Saccharomycotina yeasts ($n = 12$). The panels are designated as follows: (**A**) plant; (**B**) animal; (**C**) Pezizomycotina; (**D**) Saccharomycotina yeast.

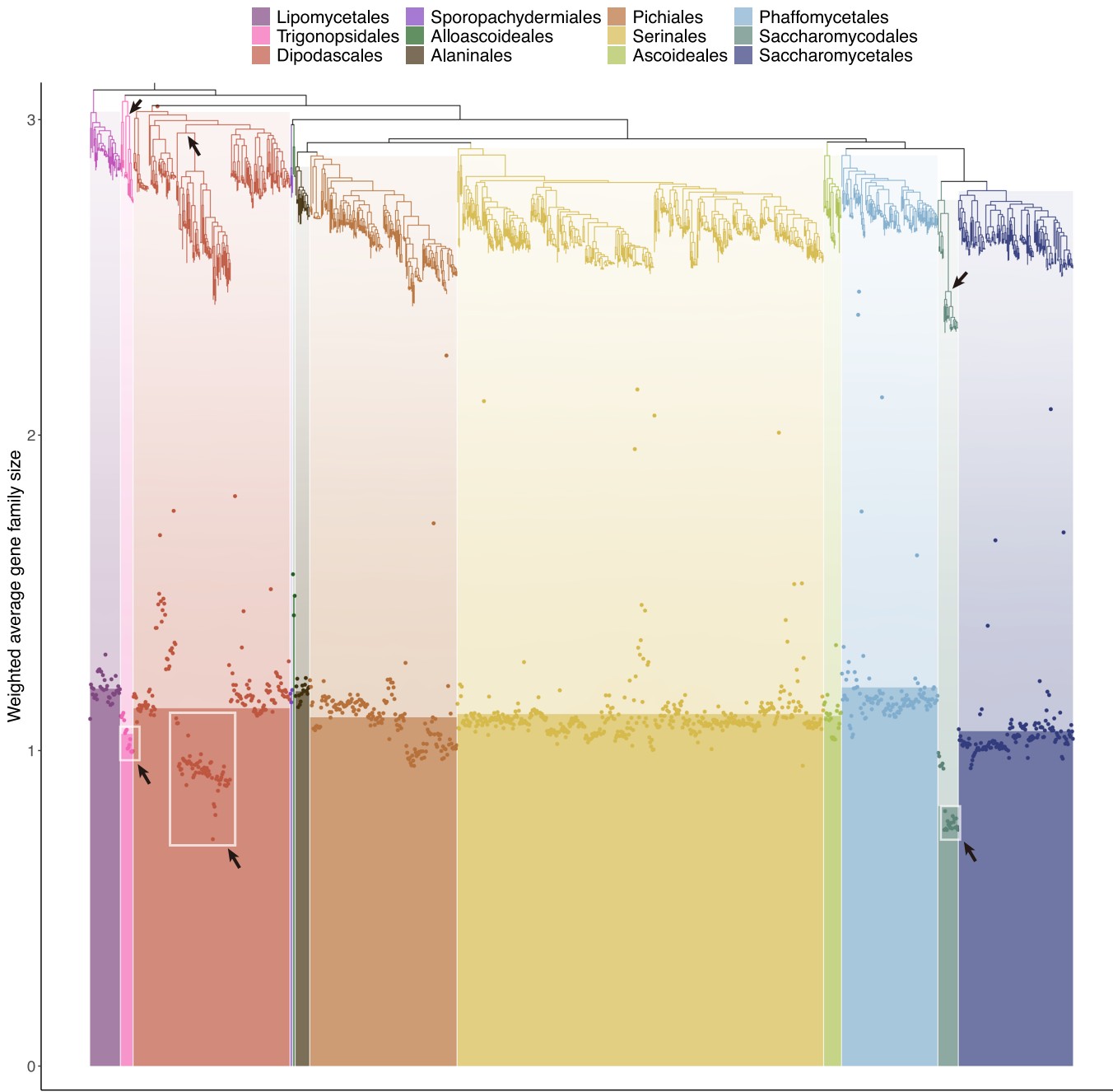

**Figure EV2. Weighted average gene family sizes across 12 orders.**

The scatter plot arranges yeast species in accordance with their placement on the phylogenetic tree, with the rectangle width depicting the number of genomes within each order. The darker color in each rectangle signifies the average of weighted average gene family size within that order, and the lighter color aligns the rectangle with its specific location on the phylogenetic tree. From left to right, the orders are represented as follows: Lipomycetales ($n = 36$), Trigonopsidales ($n = 15$), Dipodascales ($n = 184$), Sporopachydermiales ($n = 3$), Alloascoideales ($n = 3$), Ascoideales ($n = 21$), Alaninales ($n = 17$), Pichiales ($n = 173$), Serinales ($n = 430$), Phaffomycetales ($n = 113$), Saccharomycodales ($n = 24$), and Saccharomycetales ($n = 135$). A white box and arrow highlight the FELs in Trigonopsidales, Dipodascales, and Saccharomycodales.

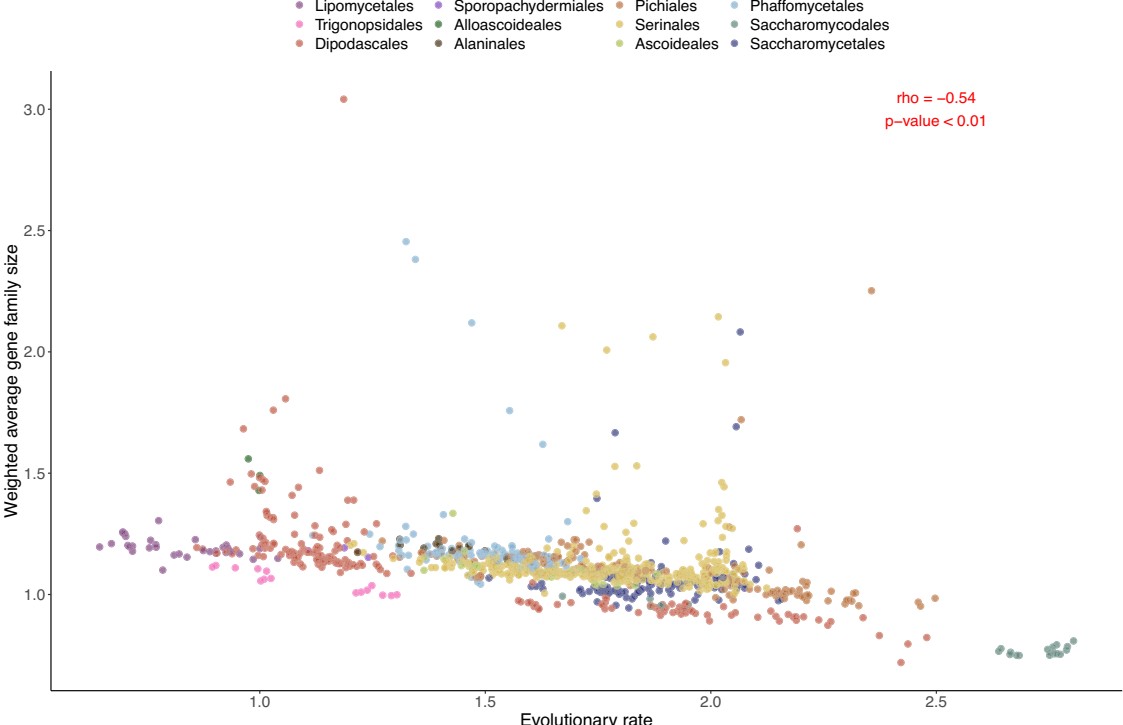

**Figure EV3. Comparative analyses of weighted average gene family sizes versus evolutionary rates in FEL and SEL gene families.**

The evolutionary rate was calculated using the branch length from the tip to the root in the phylogenetic tree. A Spearman test was conducted to assess the correlation between the weighted average size and evolutionary rate across 1154 yeasts (rho = −0.54, $P < 2.2 \times 10^{-16}$).

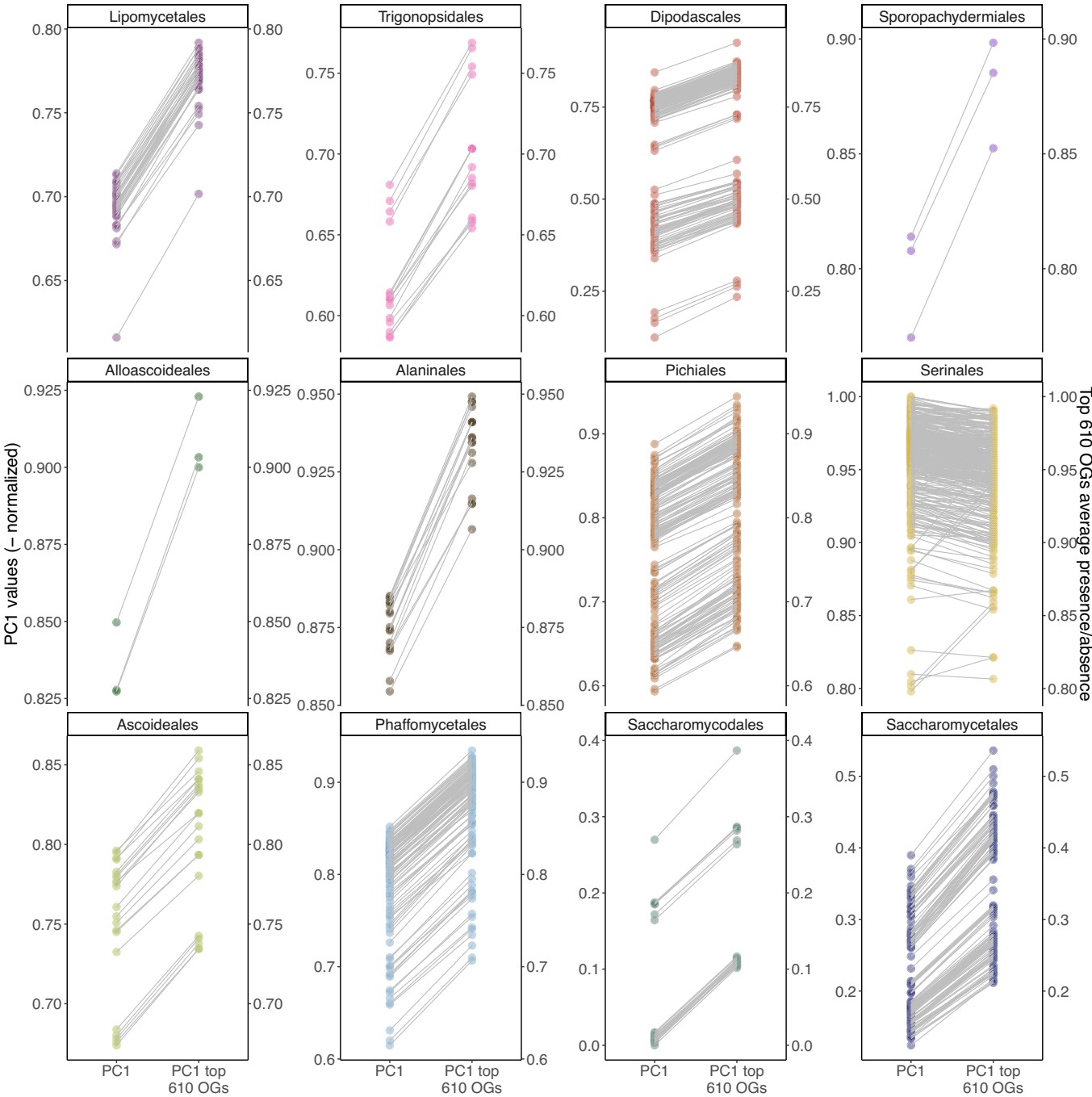

**Figure EV4.  Identification of 610 gene families representing PC1 in the PCA with a 0.5 coverage threshold.**

The correlation analysis assessed the connection between PC1 coordinates and the average presence and absence data for the top 610 gene families, which exhibited the highest absolute correlation (rho = −0.99) via the Spearman test. PC1 coordinates were reversed and then normalized. Points on the plot represent individual yeasts, with lines connecting the same point.

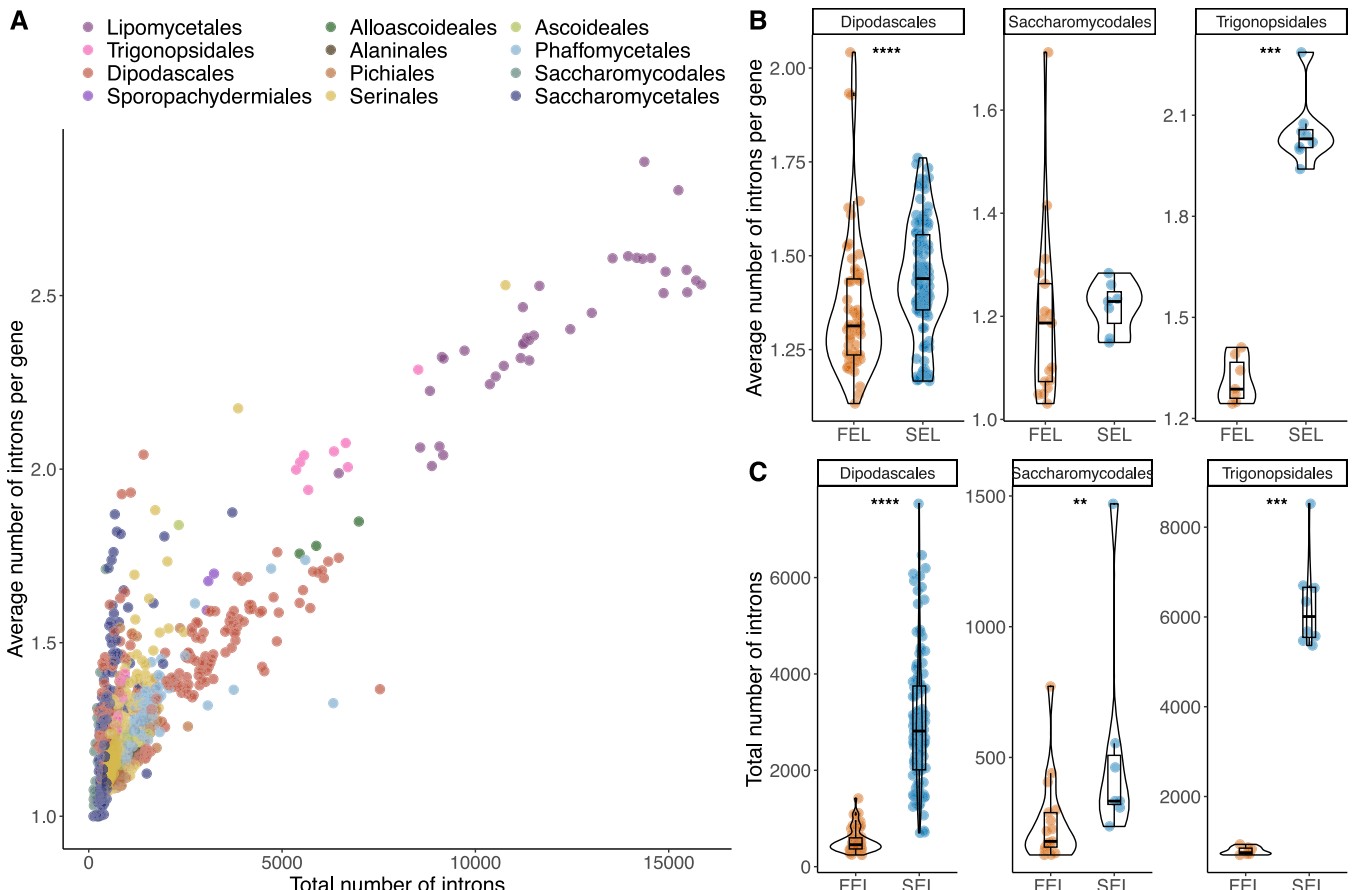

**Figure EV5. Comparative analysis of intron number between FELs and SELs.**

(A) Average intron number per gene and total number of introns across 1154 yeast species. (B) Comparison of average intron number per gene between FEL and SEL yeasts. Wilcoxon tests were used to determine significance, where "**" represents $P < 0.01$, "***" denotes $P < 0.001$, "****" indicates $P < 0.0001$. Exact p-values: Dipodascales, $P = 2.02 \times 10^{-4}$; Saccharomycodales, $P = 0.29$; Trigonopsidales $P = 3.11 \times 10^{-4}$. The center of each box plot represents the median performance, the box boundaries correspond to the upper and lower quartiles, and the whiskers extend to the 5th and 95th percentiles. The sample sizes ($n$) are as follows: for Dipodascales, FEL = 61 and SEL = 123; for Saccharomycodales, FEL = 17 and SEL = 7; for Trigonopsidales, FEL = 7 and SEL = 8. (C) Comparison of the total number of introns between FEL and SEL yeasts. Wilcoxon tests were used to determine significance, where "**" represents $P < 0.01$, "***" denotes $P < 0.001$, "****" indicates $P < 0.0001$. Exact p-values: Dipodascales, $P < 2.2 \times 10^{-16}$; Saccharomycodales, $P = 7.63 \times 10^{-3}$; Trigonopsidales $P = 3.11 \times 10^{-4}$. The center of each box plot represents the median performance, the box boundaries correspond to the upper and lower quartiles, and the whiskers extend to the 5th and 95th percentiles. The sample sizes ($n$) are as follows: for Dipodascales, FEL = 61 and SEL = 123; for Saccharomycodales, FEL = 17 and SEL = 7; for Trigonopsidales, FEL = 7 and SEL = 8.

