## [Peer Review File · Molecular Systems Biology]

Unique trajectory of gene family evolution from genomic analysis of nearly all known species in an ancient yeast lineage

Bo Feng, Yonglin Li, Biyang Xu, Hongyue Liu, Jacob Steenwyk, Kyle David, Xiaolin Tian, Carla Goncalves, Dana Opulente, Abigail LaBella, Marie-Claire Harrison, John Wolters, Shengyuan Shao, Zhaohao Chen, Kaitlin Fisher, Marizeth Groenewald, Chris Hittinger, Xing-Xing Shen, Shengying Li, Antonis Rokas, Xiaofan Zhou, and Yuanning Li

Corresponding author(s): Antonis Rokas (antonis.rokas@vanderbilt.edu), Xiaofan Zhou (xiaofan_zhou@scau.edu.cn), Yuanning Li (yuanning.li@email.sdu.edu.cn)

Review Timeline:

Submission Date:	16th Oct 24
Editorial Decision:	29th Nov 24
Revision Received:	23rd Feb 25
Editorial Decision:	5th Apr 25
Revision Received:	11th Apr 25
Accepted:	30th Apr 25

Editor: Yehu Moran

Transaction Report:

29th Nov 2024

Manuscript Number: MSB-2024-12697

Title: Unique trajectory of gene family evolution in an ancient yeast lineage

Author: Bo Feng

Yonglin Li

Hongyue Liu

Jacob Steenwyk

Kyle David

Xiaolin Tian

Biyang Xu

Carla Goncalves

Dana Opulente

Abigail LaBella

Marie-Claire Harrison

John Wolters

Shengyuan Shao

Zhaohao Chen

Kaitlin Fisher

Marizeth Groenewald

Chris Hittinger

Xing-Xing Shen

Antonis Rokas

Xiaofan Zhou

Yuanning Li

Dear Professor Rokas,

Thank you very much for submitting your work to Molecular Systems Biology. We have now heard back from the three referees who agreed to evaluate your manuscript. As you will see from the reports below, all three referees find the topic of your study of potential interest. They raise, however, some concerns on your work, which preclude its publication in its present form and in our opinion require a major revision.

When you resubmit your manuscript, please download our CHECKLIST (<https://bit.ly/EMBOPressAuthorChecklist>) and include the completed form in your submission.

Please note that the Author Checklist will be published alongside the paper as part of the transparent process (<https://www.embopress.org/page/journal/17444292/authorguide#transparentprocess>).

If you feel you can satisfactorily deal with these points and those listed by the referees, you are very welcome to submit a revised version of your manuscript. Please attach a covering letter giving details of the way in which you have handled each of the points raised by the referees. A revised manuscript will be once again subject to review and you probably understand that we can give you no guarantee at this stage that the eventual outcome will be favorable. Yet in light of the interest in the topic expressed by the three referees we are quite optimistic.

Yours sincerely,

Yehu Moran

Academic Editor

Molecular Systems Biology

We realize that it is difficult to revise to a specific deadline. In the interest of protecting the conceptual advance provided by the work, we recommend a revision within 3 months (27th Feb 2025). Please discuss the revision progress ahead of this time with the editor if you require more time to complete the revisions. Use the link below to submit your revision:

IMPORTANT: When you send your revision, we will require the following items:

1. the manuscript text in LaTeX, RTF or MS Word format
2. a letter with a detailed description of the changes made in response to the referees. Please specify clearly the exact places in the text (pages and paragraphs) where each change has been made in response to each specific comment given
3. three to four 'bullet points' highlighting the main findings of your study
4. a short 'blurb' text summarizing in two sentences the study (max. 250 characters)
5. a 'thumbnail image' (550px width and max 400px height, Illustrator, PowerPoint or jpeg format), which can be used as 'visual title' for the synopsis section of your paper.
6. Please include an author contributions statement after the Acknowledgements section (see <https://www.embopress.org/page/journal/17444292/authorguide>)
7. Please complete the CHECKLIST available at (<https://bit.ly/EMBOPressAuthorChecklist>). Please note that the Author Checklist will be published alongside the paper as part of the transparent process (<https://www.embopress.org/page/journal/17444292/authorguide#transparentprocess>).
8. When assembling figures, please refer to our figure preparation guideline in order to ensure proper formatting and readability in print as well as on screen: <https://bit.ly/EMBOPressFigurePreparationGuideline>
See also figure legend guidelines: <https://www.embopress.org/page/journal/17444292/authorguide#figureformat>
9. Please note that corresponding authors are required to supply an ORCID ID for their name upon submission of a revised manuscript (EMBO Press signed a joint statement to encourage ORCID adoption). (<https://www.embopress.org/page/journal/17444292/authorguide#editorialprocess>)
Currently, our records indicate that the ORCID for your account is 0000-0002-7248-6551.

Link Not Available

11. Include a Reagents and Tools Table as part of the Methods section, which can be downloaded from our author guidelines (<https://www.embopress.org/page/journal/17444292/authorguide#structuredmethods>)

*** PLEASE NOTE *** As part of the EMBO Press transparent editorial process initiative (see our Editorial at <https://dx.doi.org/10.1038/msb.2010.72>), Molecular Systems Biology publishes online a Review Process File with each accepted manuscripts. This file will be published in conjunction with your paper and will include the anonymous referee reports, your point-by-point response and all pertinent correspondence relating to the manuscript. If you do NOT want this File to be published, please inform the editorial office at msb@embo.org within 14 days upon receipt of the present letter.

Reviewer #1:

Reviewer report:

Unique trajectory of gene family evolution in an ancient yeast lineage

Summary:

This study by Feng et al provides novel insights into the dynamics of gene evolution across the yeast subphylum Saccharomycotina. They carry out a range of computational analyses to understand the rate of gene gain and loss both across the subphylum and between the orders within Saccharomycotina. This combined approach gives interesting insights into the role of gene gain and loss in the early evolution of the subphylum, as well as the evolution of specific traits within each order. In particular, the findings relating to higher rates of sequence evolution and increased gene loss are of interest and provide insights which lead to testable hypotheses later on (e.g. assessing the role of gene expression, carrying out functional genomics).

General remarks:

This is an important study which highlights the role of gene evolution in an important subphylum of yeast. There has been a significant amount of work on the phylogenomics and genomics of this group by the authors and this study marks a significant advance in understanding the biology of Saccharomycotina. I commend the authors for robust and inventive computational analyses. As we see significant increases in genome and proteome data across many of the major groups in eukaryotes, we are faced with new computational challenges where previous tools or algorithms built to handle data from tens of species simply do

not work when applied to hundreds or thousands of species. This study sets and standard for how to handle these large datasets and provides a useful workflow which other researchers can follow. Proof of the usefulness of these analyses are found in the myriad of interesting biological findings in this study.

Major points:

The analyses are generally of high quality, extensive, and well laid out in the manuscript. I have just a few points to be addressed.

(1) Orphan genes are given very little attention, however, represent an interesting set of genes (which have been extensively studied in some yeast lineages). I wonder if the authors could expand more on the potential mechanisms of origin for the orphan genes gained in the ancestor of each order. Do they share any homology outside of their respective orders i.e. are they the result of duplication followed by rapid sequence divergence or did they originate through de novo gene genesis? Searching for the presence of domains, or carrying out a search against known genes, will give insight into this.

(2) Saccharomycodales does not show strong evidence for lower intron counts between FEL and SEL, compared to the other two orders. Yet you state that 50% of the genes associated with the splicing pathway are lost in Saccharomycodales. Conversely, Trigonopsidales shows very strong evidence for lower intron counts in FEL, however, have no significant losses of genes related to this pathway.

This is a contradiction which should be highlighted in the Discussion as it flags a broader question - how often is gene loss associated with a given trait, versus how often is gene loss random, without effect. In other words, how often is gene loss (i) adaptive, (ii) have a negative fitness affect, or (iii) has no effect and is a result of drift?

Does loss of intron result in mRNA splicing gene loss because they're not required any more (i.e. use it or lose it), or does gene loss lead to intron loss?

I'm sure there are other traits which will correlate with mRNA splicing gene loss other than intron loss (as mentioned in the Discussion, related to generating novel transcript variants), and I also know that this is a difficult question to answer, however I think warrants more exploration in the Discussion. There is extensive literature on the adaptive role of gene loss which should be addressed (e.g. work by Cristian Cañestro to name just one).

(3) Parts of the Discussion could be improved/rewritten to provide more context for the importance of this study. In particular, the last paragraph reads like a list of analyses carried out and does not provide much insight. I think the authors are trying to highlight the types of analyses possible when analysing thousands of genomes, however it seems quite redundant. I would recommend cutting down this last paragraph to the key points and merging it with the previous one.

Minor points:

(1) Line 187: rho value stated in main text (-0.41) differs from that shown on Figure S3 (-0.54).

(2) Line 193: Trigonopsidales stated as having lineage-specific variations in weighted average gene family sizes, however in Supplementary table S3 p-value is 0.230157621408895 for Trigonopsidales? While it does look like there is significance in evol. rate and weighted gene family size in this order in Figure 2g&j, this inconsistency should be corrected.

(3) Lines 194: Points to Figure S4, however this shows information for two orders (Pichiales and Serinales) not mentioned in this sentence (which refers to Dipodascales, Saccharomycodales, and Trigonopsidales)? Are these shown as examples where evol rate and weighted gene family size do not correlate?

(4) Line 217: Refers to Figure 2k-m which do not exist, I believe this should refer to Figure 3a.

(5) Line 258: (Figure 4c) should be (Figure 3c)

(6) I assume that the x-axis in Figure 4a is ordered according to the species phylogeny? If it isn't, then it should be. If it is then it might be worth adding a comment that this is the case in the Figure legend.

(7) Line 458: I would suggest change 'occurred at the base of the animal and plant phylogenies' to 'occurred multiple times across the animal and plant phylogenies'. WGD events in animals did not occur at the base of the phylogeny, but rather in vertebrates (which is an old WGD event, but not the base of the tree) multiple times, and in smaller more recent clades (e.g. arachnids, earthworms etc).

Reviewer #2:

Feng and colleagues analysed gene family evolution across a comprehensive set of yeast genomes (N=1154). Comparisons with plant, animal, and filamentous ascomycetes lineages are reported together with detailed analyses across the specific yeast orders. The first part of the manuscript focuses on the relationships between gene family size and (i) gene content (ii) genome

size among groups, with the conclusion that yeasts tend to have larger gene families/gene number similarly to plants. This is a very interesting observation. Next, evolutionary rates were correlated to gene family size between and within yeast orders. Faster evolving lineages (small size, fast rate) and slower evolving lineages (larger gene size and slower rate) were scattered across the yeast tree within each order. The contraction in RELs was associated with loss of genes related to RNA splicing, cell division and metabolism. The order Dipodascales was further dissected as an extreme case. While losses represent a general trend, waves of duplication and loss better describe the observed pattern of genome evolution.

General remarks

Overall, I greatly enjoyed the manuscript, and I think it is suitable for publication on MSB with some changes listed below. The fact that these analyses were leveraged to such large dataset of fungi is a key advancement of this manuscript.

Major points

What I missed was a deeper discussion of the results on the "Uniqueness" that is mentioned in the title. Is the uniqueness in the fact that a contraction was detected in the SCA to the MRCA followed by waves of duplications and loss? How is it different (following the "Unique" adjective) from the gene loss, turnover and several large duplication events detected in pre-fungal lineages? How does it differ from other eukaryotic lineages?

In general Fungi represent a genomically non-uniform assemblage of species, could you expand the discussion on gradual vs. episodic burst of contractions/duplications in the discussion?

I also find misleading this sentence in the abstract: "We found that yeast gene family and genome evolution are distinct from plants, animals, and filamentous ascomycetes and are characterized by small genome sizes and smaller gene numbers but larger gene family sizes".

1) I do not think the genome size statement is supported in the current text, except for the weak correlation identified between gene file size and genome size.

2) Comparing organism with equivalent number of protein coding genes yeasts displayed similar weighted average sizes as plants but larger than animals and filamentous ascomycetes. (cit. line 156). The current statement in the abstract is overamplifying the results.

Next general question: did you correct/control/account for horizontal gene transfer at any point of the study? How could it affect this analysis?

Specific comments:

- Line 130. For clarity it would be interesting to specify better in the introduction known patterns in filamentous ascomycetes, animals, and plants. In animals and plants some of the patterns are briefly mentioned at the beginning but in a rather confusing way (e.g. was it a gradual expansion/abrupt, accompanied by gene loss and on). Known patterns in filamentous ascomycetes are never mentioned. Please highlight better in the text if this is one of the first study for this specific comparison.
- Line 156: could you better define the range of "equivalent"?
- Line 157: yeasts displayed similar weighted average sizes to plants and larger sizes than filamentous ascomycetes and animals (Figure 1b). I wonder if there is a better way to highlight this considering the overlapping points. Could you overlay yeast to plants and plants to yeast/further increase the alpha factor?
- Line 217. I am unable to find the panels 2k-m in Figure 2. If the absence is intentional please adjust the text.
- Line 234. Figure legends and citations in the text might be inconsistent. First, please define better the 0.5 coverage threshold. Next, it feels like the 0.1 and 0.5 values go back and forth. In Figure S1 the threshold (red line) is at 0.1 but Figure S1 is cited for threshold 0.5. The main text Figure 3 has a PCA with coverage 0.1. Figure S6 on the other hand shows the threshold at 0.5 but the main text cites both figures as having 0.1. I would suggest putting the final threshold you used in the main text figure and the more relaxed one in a single supplementary figure. Justify better why 0.5 was used. Also check which threshold was described in the methods Line 621.
- Line 253: PCA scatter plot (cite the specific panel you refer to).
- Line 620: connected to the previous point you mentioned in the methods citation 3 to support the use of PCA for the presence/absence analysis. The reference cited used a principal coordinate analysis (PCoA). Could you comment on the suitability of the 2 methods for your purpose? Also, a phylogenetic-aware analysis would have changed anything?
- Line 318: in Figure 4d the difference in long and short dashes is not visible immediately.
- Line 372: reference Figure 5.
- Line 407: "specifically" is misplaced? Please rephrase the sentence for clarity.
- Line 404: what are the potential implications in this last paragraph? Only a small portion of gains include novelty, and we are rather observing expansion of ancestral families?
- Line 417: the more specialised lifestyle/metabolic capacity is not so universal as this sentence hints to but has been identified specifically only in Dipodascales. When looking for more details regarding this aspect in other orders I can only find line 382 and a generic "gene families involved in metabolism". It would be interesting to expand, elaborate this part in order to corroborate the conclusion/statement in line 417.
- Line 527: could you annotate in e.g. supplementary table 1 which species were used as -core in orthofinder analysis?

- Line 544: can you specify the availability of this script or add it to the repository if not available yet? Same for the other R custom script in Line 632.

Reviewer #3:

In this manuscript, the authors examined gene family evolution in the Saccharomycotina and find that these yeasts have smaller gene numbers and smaller weighted average gene family sizes than plants, animals and filamentous ascomycetes (Pezizomycotina), but that among animals, plants and Saccharomycotina with comparable gene numbers, the Saccharomycotina exhibit larger weighted average gene family sizes. Furthermore, faster-evolving Saccharomycotina lineages are found to exhibit higher speciation rates and higher rates of gene loss, with e.g. mRNA splicing and carbohydrate metabolism genes enriched among the lost genes. Overall, I think this an interesting study but there are a number of methodological aspects that I have questions about.

Major comments

- Half of the introduction is actually a summary of the main results of the analyses, using terms such as the weighted average gene family size that are only introduced later on in the results section. This summary overlaps substantially with other sections of the manuscript and should be shortened in my opinion.
- The weighted average gene family size measure is central to the analyses performed in the manuscript, but is not defined adequately in the results or methods sections (and also not in the 1K plant transcriptome paper where it has been used before). As I understand it, a weighted sum is made of the sizes of all gene families in some species, e.g. a Saccharomycotina yeast (although the description in the methods suggests the measure is calculated for a set of species as a whole), with the weight for a gene family being the inverse of the maximum copy number of that gene family in all Saccharomycotina. It is not clear why such weighting is necessary and what its effects are. The weighting scheme also relies heavily on the maximum gene family sizes in the set of species investigated, which are at risk of being outlier values due to e.g. suboptimal genome assemblies or annotations. It is also not clear to me why the resulting weighted averages would be comparable across different kingdoms, as species of different kingdoms would not be compared on equal footing because of the use of entirely different weighting schemes. In theory one badly assembled genome with inflated gene family sizes could be sufficient to skew the weighted averages of an entire kingdom.
- Related to this, it is unclear to me whether for the analyses across 12 yeast orders the weights have been calculated per order or for all orders simultaneously.
- On several occasions, conflicting statements are made in different parts of the text. E.g. the abstract states that yeasts are 'characterized by small genome sizes and smaller gene numbers but larger gene family sizes' while in the introduction it is stated that (line 113-114) 'yeasts have smaller weighted average gene family sizes due to fewer gene counts'. This needs to be clarified: it is the ratio of average gene family size to gene number that is larger in Saccharomycotina than in other clades.
- Similarly, the authors state on line 433 that 'These gene family contractions and losses in FELs may contribute to their higher evolutionary and speciation rates (Figures 2e-f and 3e) by enabling rapid genomic adaptations that optimize cellular processes crucial for survival and reproduction in diverse and challenging environments.', while on line 467 it is stated that 'when we control for gene number, we found that yeasts exhibit larger weighted average gene family sizes than both filamentous ascomycetes and animals and are on par with plants (Figure 1b). Larger gene family sizes may provide redundancy and adaptability in biological processes'. These statements seem to be at odds with each other. While gene losses may indeed conceivably lead to higher speciation rates if there is differential gene loss in different subpopulations leading to reproductive isolation, I don't see the link between increased gene loss and increased survival or adaptability.
- The authors base their gene family expansion and contraction statements in FEL versus SEL yeasts on fold changes in gene family size, equating e.g. fold changes >1.5 in FEL versus SEL yeasts as expansions in the FEL lineage. Couldn't these equally likely be contractions in the SEL lineage instead? Why didn't the authors use e.g. CAFE here to assess expansions and contractions, as they do further on in the manuscript? Without reference to gene family sizes in the most recent common ancestor, any assessment of expansion or contraction in a particular lineage seems arbitrary.
- The authors use PCA analysis on binary presence-absence data for gene families and afterwards identify the gene families that contributed most to the first two PCs. Using PCA on binary data is however not advised for anything else than pure dimensionality reduction (i.e. not for interpretation of variable loadings on the components), see e.g. Collins, M., Dasgupta, S., & Schapire, R. E. (2002). A generalization of principal components analysis to the exponential family. *Advances in neural information processing systems* pp. 617-624. Using PCA on the full gene family size data rather than presence-absence would have been better although still not ideal.

Minor comments

- Line 102 and other instances: 'yeast' is not synonymous to 'Saccharomycotina'
- Line 212: '... for each gene family and each pair': it is not clear what the pairs are here.
- Line 646: 'species with larger coordinates on the PC tend to have more gene copies in the top i gene families': I don't see why this should be the case, as the number of gene copies was not taken into account in the PCA analysis.

Response to Reviewers

Reviewer #1:

Reviewer report:

Unique trajectory of gene family evolution in an ancient yeast lineage

Summary:

R1.1 This study by Feng et al provides novel insights into the dynamics of gene evolution across the yeast subphylum Saccharomycotina. They carry out a range of computational analyses to understand the rate of gene gain and loss both across the subphylum and between the orders within Saccharomycotina. This combined approach gives interesting insights into the role of gene gain and loss in the early evolution of the subphylum, as well as the evolution of specific traits within each order. In particular, the findings relating to higher rates of sequence evolution and increased gene loss are of interest and provide insights which lead to testable hypotheses later on (e.g. assessing the role of gene expression, carrying out functional genomics).

AUTHORS' RESPONSE: We thank the reviewer for appreciating the significance of our work and for the constructive feedback.

General remarks:

R1.2 This is an important study which highlights the role of gene evolution in an important subphylum of yeast. There has been a significant amount of work on the phylogenomics and genomics of this group by the authors and this study marks a significant advance in understanding the biology of Saccharomycotina. I commend the authors for robust and inventive computational analyses. As we see significant increases in genome and proteome data across many of the major groups in eukaryotes, we are faced with new computational challenges where previous tools or algorithms built to handle data from tens of species simply do not work when applied to hundreds or thousands of species. This study sets and standard for how to handle these large datasets and provides a useful workflow which other researchers can follow. Proof of the usefulness of these analyses are found in the myriad of interesting biological findings in this study.

AUTHORS' RESPONSE: We are grateful for your kind remarks on the importance of our study and the computational framework we have developed. It is highly encouraging to hear that you view our approach as a standard for handling large-scale datasets.

Major points:

The analyses are generally of high quality, extensive, and well laid out in the manuscript. I have just a few points to be addressed.

R1.3 (1) Orphan genes are given very little attention, however, represent an interesting set of genes (which have been extensively studied in some yeast lineages). I wonder if the authors could expand more on the potential mechanisms of origin for the orphan genes gained in the ancestor of each order. Do they share any homology outside of their respective orders i.e. are they the result of duplication followed by rapid sequence divergence or did they originate through de novo gene genesis? Searching for the presence of domains, or carrying out a search against known genes, will give insight into this.

AUTHORS' RESPONSE: Thank you for your insightful comments on orphan genes and for emphasizing the importance of investigating their potential mechanisms of origin. We have expanded our analyses and detailed the findings in the Results and Methods sections. Specifically, we examined whether orphan genes present in the ancestor of each order show homology outside their respective orders and explored their potential origins, including duplication with rapid sequence divergence, de novo gene emergence, and horizontal gene transfer (HGT). To address these questions, we conducted two BLASTP comparisons:

1. Between orphan genes and our dataset of 1,154 yeast genomes, using thresholds based on a previous in depth study using a sequence homology search approach (Pearson 2013): e-value $< 10^{-5}$, bit score > 50 , and percent identity $> 40\%$.
2. Between orphan genes and the NCBI non-redundant (NR) protein database, with thresholds of e-value $< 10^{-5}$ and bit score > 50 .

Our workflow proceeded in the following manner. When a homologous protein was identified in search #1 outside the respective order, the orphan gene was considered to have likely originated from duplication or speciation followed by rapid sequence divergence. If a homolog was not found in search #1 but was detected in search #2, the orphan gene was inferred to have potentially originated via HGT. Finally, orphan genes without homologs in either search were categorized as putatively *de novo*.

The results revealed that 24,577 orphan genes (96.5%) appear to have emerged *de novo*, only 36 genes (0.2%) likely arose from speciation or duplication events after the most recent common ancestor of Saccharomycotina, and 865 genes (3.4%) are likely the result of HGT (Reviewer #1 Figure 1).

Reviewer #1 Figure 1. The blue circle (<1>) represents all orphan genes, the orange circle (<2>) represents orphan genes with protein homologs in other orders (excluding the order to which the specific orphan gene belongs), and the yellow circle (<3>) represents orphan genes with protein homologs in the NR database (excluding Saccharomycotina).

R1.4 (2) Saccharomycodales does not show strong evidence for lower intron counts between FEL and SEL, compared to the other two orders. Yet you state that 50% of the genes associated with the splicing pathway are lost in Saccharomycodales. Conversely, Trigonopsidales shows very strong evidence for lower intron counts in FEL, however, have no significant losses of genes related to this pathway. This is a contradiction which should be highlighted in the Discussion as it flags a broader question - how often is gene loss associated with a given trait, versus how often is gene loss random, without effect. In other words, how often is gene loss (i) adaptive, (ii) have a negative fitness affect, or (iii) has no effect and is a result of drift?

AUTHORS' RESPONSE: We appreciate your insights about the lack of correlation between gene loss and intron reduction in Saccharomycodales and Trigonopsidales, which can shed light to the broader question of whether gene loss is adaptive, neutral, or deleterious.

To address this, we have added a section in the Discussion exploring this issue. We suggest that gene loss can sometimes be linked to adaptive processes, such as reducing metabolic costs or enhancing microbial pathogenicity, and other times it may be linked to neutral drift, having minimal phenotypic impact when functional redundancy is present.

For Saccharomycodales, the average intron number across the order is much lower than in the other two orders, with approximately 1.2 introns per gene (compared to ~1.7 for Trigonopsidales and ~1.4 for Dipodascales) and around 400 total introns (compared to ~3700 for Trigonopsidales and ~2100 for Dipodascales) (Figure S10B and C (Figure EV5B and C)). As a result, the loss of splicing pathway genes in Saccharomycodales does not significantly affect intron numbers, likely because the substantial loss of introns occurred first. In contrast, Trigonopsidales have slightly higher intron counts, averaging 1.3 introns per gene in FEL and 2.0 in SEL, with over 1,000 total introns in FEL and 6,000 in SEL. Consequently, even minor gene losses in Trigonopsidales can lead to substantial reductions in intron numbers.

We emphasize that gene loss and functional phenotypes are influenced by multiple factors and are not always correlated. Future studies involving larger datasets and experimental validation will further clarify the balance between adaptive and neutral effects of gene loss. Thank you for highlighting this critical aspect, which has deepened our understanding of genome evolution.

R1.5 Does loss of intron result in mRNA splicing gene loss because they're not required any more (i.e. use it or lose it), or does gene loss lead to intron loss? I'm sure there are other traits which will correlate with mRNA splicing gene loss other than intron loss (as mentioned in the Discussion, related to generating novel transcript variants), and I also know that this is a difficult question to answer, however I think warrants more exploration in the Discussion. There is extensive literature on the adaptive role of gene loss which should be addressed (e.g. work by Cristian Cañestro to name just one).

AUTHORS' RESPONSE: We thank you for your insightful comments on the causal relationship between intron loss and mRNA splicing gene loss, as well as the adaptive role of gene loss. As you emphasized, this is a critical yet unresolved question in evolutionary biology: do trait changes occur first, making certain genes redundant (“use it or lose it”), or does the loss of key genes drive subsequent trait changes?

To address the Reviewer’s comment, we have expanded the Discussion to explore possible roles of gene loss in evolution and adaptation, including:

1. Adaptive loss: Splicing gene loss might confer advantages by reducing energy or resource demands in specific contexts.
2. Functional impacts: Beyond intron number, splicing gene loss might affect transcript diversity by reducing splicing variability.

We have also added references, including Cristian Cañestro’s work, to provide more context on the evolutionary significance of gene loss. Although the order of intron

and gene loss remains unclear due to limited functional evidence, future genomics and experimental studies could clarify these dynamics.

R1.6 (3) Parts of the Discussion could be improved/rewritten to provide more context for the importance of this study. In particular, the last paragraph reads like a list of analyses carried out and does not provide much insight. I think the authors are trying to highlight the types of analyses possible when analysing thousands of genomes, however it seems quite redundant. I would recommend cutting down this last paragraph to the key points and merging it with the previous one.

AUTHORS' RESPONSE: Thank you for your insightful feedback on the Discussion section. We recognize that the final paragraph was repetitive and lacked a strong focus on the broader implications of our findings, and we have made substantial revisions to address these issues.

In the revised manuscript, the final paragraph was rewritten to emphasize the key insights gained from our analyses and their significance in understanding large-scale genomic data, highlighting their relevance to the research community. To improve the flow of ideas, we merged this paragraph with the previous one, eliminating redundancies and creating a more logical progression that connects our findings to their broader implications. Additionally, we have added context to underscore the importance of our results, elaborating on how they can guide future research and applications in evolutionary biology and comparative genomics, thereby enhancing the relevance of our study.

We believe these changes make the Discussion more cohesive as you suggested. Thank you again for your constructive suggestions, which have greatly improved the quality of our manuscript.

Minor points:

R1.7 (1) Line 187: rho value stated in main text (-0.41) differs from that shown on Figure S3 (-0.54).

AUTHORS' RESPONSE: Fixed as suggested.

R1.8 (2) Line 193: Trigonopsidales stated as having lineage-specific variations in weighted average gene family sizes, however in Supplementary table S3 p-value is 0.230157621408895 for Trigonopsidales? While it does look like there is significance in evol. rate and weighted gene family size in this order in Figure 2g&j, this inconsistency should be corrected.

AUTHORS' RESPONSE: Thank you for pointing this out. We acknowledge that the p-value for the multimodality test of weighted average gene family size in Trigonopsidales is larger than 0.05. However, the value (0.22) is considerably

smaller than the p-values of other orders (> 0.75), which may still indicate notable differences in the distribution of gene family sizes for this order. We have updated the expression in the Results section to more accurately reflect this observation.

R1.9 (3) Lines 194: Points to Figure S4, however this shows information for two orders (Pichiales and Serinales) not mentioned in this sentence (which refers to Dipodascales, Saccharomycodales, and Trigonopsidales)? Are these shown as examples where evol rate and weighted gene family size do not correlate?

AUTHORS' RESPONSE: Thank you for your comment. Pichiales and Serinales were included in the analysis because they showed significant differences in branch length in the previous multimodality test. When mapping branch lengths and weighted average gene family sizes onto the tree, these two orders provided important examples, revealing that their evolutionary rates and weighted average gene family sizes are not correlated. To ensure clarity, we have revised the sentence to better align with the multimodality test results, minimizing potential confusion for readers.

R1.10 (4) Line 217: Refers to Figure 2k-m which do not exist, I believe this should refer to Figure 3a.

AUTHORS' RESPONSE: Corrected as suggested.

R1.11 (5) Line 258: (Figure 4c) should be (Figure 3c)

AUTHORS' RESPONSE: Corrected as suggested.

R1.12 (6) I assume that the x-axis in Figure 4a is ordered according to the species phylogeny? If it isn't, then it should be. If it is then it might be worth adding a comment that this is the case in the Figure legend.

AUTHORS' RESPONSE: Thank you for pointing this out. The x-axis in Figure 4a was not fully aligned with the species phylogeny. We have revised the figure to ensure proper ordering and added a note in the legend to clarify this for improved reader understanding.

R1.13 (7) Line 458: I would suggest change 'occurred at the base of the animal and plant phylogenies' to 'occurred multiple times across the animal and plant phylogenies'. WGD events in animals did not occur at the base of the phylogeny, but rather in vertebrates (which is an old WGD event, but not the base of the tree) multiple times, and in smaller more recent clades (e.g. arachnids, earthworms etc).

AUTHORS' RESPONSE: Thank you for the suggestion. We agree with your point and have revised the sentence to state: "WGD events occurred multiple times across the animal and plant phylogenies."

Reviewer #2:

R2.1 Feng and colleagues analysed gene family evolution across a comprehensive set of yeast genomes (N=1154). Comparisons with plant, animal, and filamentous ascomycetes lineages are reported together with detailed analyses across the specific yeast orders. The first part of the manuscript focuses on the relationships between gene family size and (i) gene content (ii) genome size among groups, with the conclusion that yeasts tend to have larger gene families/gene number similarly to plants. This is a very interesting observation. Next, evolutionary rates were correlated to gene family size between and within yeast orders. Faster evolving lineages (small size, fast rate) and slower evolving lineages (larger gene size and slower rate) were scattered across the yeast tree within each order. The contraction in RELs was associated with loss of genes related to RNA splicing, cell division and metabolism. The order Dipodascales was further dissected as an extreme case. While losses represent a general trend, waves of duplication and loss better describe the observed pattern of genome evolution.

General remarks:

Overall, I greatly enjoyed the manuscript, and I think it is suitable for publication on MSB with some changes listed below. The fact that these analyses were leveraged to such large dataset of fungi is a key advancement of this manuscript.

AUTHORS' RESPONSE: We are delighted to hear that you enjoyed the study and consider it suitable for publication in *Molecular Systems Biology* with some revisions. We also appreciate your recognition of the significance of leveraging such a large dataset of fungi in our analyses. We have carefully addressed the changes you suggested and have made the necessary revisions to further enhance our manuscript.

Major points:

R2.2 What I missed was a deeper discussion of the results on the "Uniqueness" that is mentioned in the title. Is the uniqueness in the fact that a contraction was detected in the SCA to the MRCA followed by waves of duplications and loss? How is it different (following the "Unique" adjective) from the gene loss, turnover and several large duplication events detected in pre-fungal lineages? How does it differ from other eukaryotic lineages?

AUTHORS' RESPONSE: Thank you for emphasizing the need to elaborate on the 'Uniqueness' in our title. We agree that further clarification is essential.

In the revised manuscript, we added a dedicated section in the Discussion to detail why the SCA exhibits a unique evolutionary trajectory. The uniqueness of the gene family evolution in yeasts, as we observed, is reflected in the pattern of gene family contraction in the ancestral yeast lineages, particularly within the SCA leading to the

MRCA. This contraction is followed by waves of gene duplications and losses, a pattern that contrasts sharply with the more common gene duplication and turnover events seen in other eukaryotic lineages, especially from the last universal fungal ancestor (LUFA) to Dikarya.

For example, while major gene duplications, such as whole-genome duplications (WGD) and tandem duplications, played critical roles in shaping the gene repertoires of many other fungal phyla (e.g., chytrids and zygomycetes) and eukaryotes (e.g., plants and animals), the ancestral yeasts primarily experienced gene losses. This is evident in the comparisons of FELs vs. SELs (Figure 3) and several ancestral branches near the root of the yeast phylogeny (Figure 5). This suggests a distinct evolutionary trajectory for yeasts, differing from the broader eukaryotic context where duplications and expansions have often driven functional diversification. The widespread gene loss we report in yeasts (Shen et al., 2018), along with the shift from metabolic generalism to specialism (Opulente et al., 2024), likely represents a continuation of early fungal evolutionary patterns, where the duplication of metabolism- and transport-related genes laid the foundation for the high metabolic diversity of the Saccharomycotina subphylum.

With these additions, we aim to provide readers with a clearer understanding of the evolutionary processes that set the SCA apart from other fungal and non-fungal lineages.

R2.3 In general Fungi represent a genomically non-uniform assemblage of species, could you expand the discussion on gradual vs. episodic burst of contractions/duplications in the discussion?

AUTHORS' RESPONSE: We appreciate your thoughtful suggestion. In the revised manuscript, we have expanded the Discussion to provide a more detailed comparison of gradual versus episodic bursts of gene family contractions and duplications in fungi.

Specifically, we have:

1. Explained how gradual changes, driven by sustained ecological pressures, fine-tune metabolic and regulatory networks for environmental adaptation.
2. Discussed episodic bursts, such as whole-genome duplications and horizontal gene transfer, which drive rapid diversification and functional innovation during ecological disruptions.
3. Highlighted the interplay between gradual adaptation and episodic events, emphasizing their joint contribution to fungal genomic complexity.

These revisions aim to provide a clearer and more comprehensive explanation of how fungi, as a genomically diverse group, exhibit different patterns of gene family evolution. Your suggestion allowed us to enhance our Discussion and present a more nuanced view of fungal genomic evolution. Thank you again for your valuable feedback.

R2.4 I also find misleading this sentence in the abstract: "We found that yeast gene family and genome evolution are distinct from plants, animals, and filamentous ascomycetes and are characterized by small genome sizes and smaller gene numbers but larger gene family sizes".

1) I do not think the genome size statement is supported in the current text, except for the weak correlation identified between gene file size and genome size.

2) Comparing organism with equivalent number of protein coding genes yeasts displayed similar weighted average sizes as plants but larger than animals and filamentous ascomycetes. (cit. line 156). The current statement in the abstract is overamplifying the results.

AUTHORS' RESPONSE: Thank you for highlighting potential overstatements in the abstract. We have addressed both points as follows:

1. Support for genome size

We agree that our data show only a weak correlation between gene family size and genome size and do not conclusively demonstrate that yeasts have smaller genomes than all other lineages. We have revised the manuscript to avoid claims suggesting yeasts consistently exhibit "small genome sizes."

2. Gene family size comparison

We acknowledge that the statement about "larger gene family sizes" was overly general. Our results show that yeasts, when compared to organisms with similar numbers of protein-coding genes, exhibit weighted average gene family sizes comparable to plants and larger than those of animals and filamentous ascomycetes. We have revised the abstract to reflect this more accurately.

Below is the revised text in the Abstract:

"We found that yeast gene family evolution differs from that of plants, animals, and filamentous ascomycetes, and is characterized by smaller overall gene numbers yet larger gene family sizes for a given gene number."

R2.5 Next general question: did you correct/control/account for horizontal gene transfer at any point of the study? How could it affect this analysis?

AUTHORS' RESPONSE: We appreciate your important question about horizontal gene transfer (HGT) and its potential impact on our study. In our previous research

on yeast genome evolution, we found that yeasts exhibit very low levels of HGT, with approximately 0.04% to 0.06% of genes putatively acquired via HGT (Shen et al., 2018). To ensure that HGT does not influence the results of our current study, we excluded HGT-related gene families by using BLASTP to filter out the 878 HGT-acquired genes identified in our previous work (Shen et al., 2018). We then reran the key analyses, including fold-change and PCA, to evaluate the impact of these exclusions on our findings. The results confirmed that the exclusion of HGT-related genes did not affect the overall conclusions of the study.

Specifically, we identified 114 HGT-related gene families in the 0.1 coverage dataset (Reviewer #2 Figure 1A) used for fold-change analysis, and 66 in the 0.5 coverage dataset (Reviewer #2 Figure 1B) used for PCA. After excluding these gene families, we performed the following analyses:

1. Multimodality test

Excluding HGT genes yielded results consistent with the original dataset (Reviewer #2 Table 1). Orders such as Dipodascales and Saccharomycodales continued to show significant p-values (< 0.05), while Ascoideales, Pichiales, and Trigonopsidales displayed smaller p-values (< 0.24) compared to other orders (> 0.75).

2. Weighted average gene family size and FEL/SEL identification

The filtered 0.1 coverage dataset revealed similar weighted average gene family sizes (~ 1.12 for most yeasts) and patterns as the original (Reviewer #2 Figure 2A). Saccharomycodales still showed smaller gene family sizes. FELs and SELs were consistently identified in Trigonopsidales, Dipodascales, and Saccharomycodales (Reviewer #2 Figure 2B).

3. Fold change, PCA, and functional enrichment

Fold change analysis confirmed significant gene family contraction and loss in FELs across Saccharomycodales, Dipodascales, and Trigonopsidales, with enrichment in similar functional categories (e.g., carbon metabolism, RNA splicing) (Reviewer #2 Figure 3A and 3B). PCA results showed consistent distribution patterns, with FELs and SELs distinctly separated (Reviewer #2 Figure 3C). The 591 gene families driving differences along PC1 remained functionally enriched in DASH complex, kinetochore, and spindle attachment processes (Reviewer #2 Figure 3D).

In summary, excluding HGT-related gene families did not alter our primary patterns or conclusions, reinforcing the robustness of our results.

Reviewer #2 Figure 1. Distribution of HGT genes in 0.1 and 0.5 coverage gene family datasets. (A) Mean coverage indicates the average coverage of gene families across 12 orders of Saccharomycotina yeasts, with 114 out of 5,551 gene families containing HGT genes in the 0.1 coverage dataset. **(B)** 66 out of 4,261 gene families contain HGT genes in the 0.5 coverage dataset.

Order	P-value branch length	P-value (weighted avg. size – original)	P-value (weighted avg. size – no HGT genes)
Sporopachydermiales	1	1	1
Alloascoideales	1	1	1
Phaffomycetales	0.93557816	0.99175612	0.99983659
Alaninales	0.78693781	0.82751495	0.69063602
Saccharomycetales	0.54312995	0.75661047	0.48133129
Ascoideales	0.3626511	0.06695642	0.07597614
Lipomycetales	0.15921014	0.90150576	0.75902812
Dipodascales	0.04092472	0	0
Saccharomycodales	0.01029089	0.01857093	0.01482688
Pichiales	0.00428332	0.21538287	0.16591895
Serinales	0.0036688	0.99732383	0.99738696
Trigonopsidales	0.00081514	0.23015762	0.15117749

Reviewer #2 Table 1. Multimodality analysis of branch length and weighted average gene family size. The P-value (weighted avg. size - original) represents the multimodality test conducted on the original dataset with 0.1 coverage. The P-value (weighted avg. size - no HGT genes) refers to the test -performed on the same dataset after filtering out gene families containing HGT genes. We identified FELs and SELs in three orders: Dipodascales, Saccharomycodales, and Trigonopsidales, which are highlighted in red. P-values that are significant or significantly smaller than others are also marked in red.

Reviewer #2 Figure 2. weighted average gene family size of 0.1 coverage dataset excluding HGT genes. The arrangement of yeasts on the x-axis follows the same order as in the phylogenetic tree and aligns with Figure 2 and Figure S2 (Figure EV2) in our manuscript.

Reviewer #2 Figure 3. Fold change PCA, and functional enrichment analyses excluding HGT genes. Similar to Figure 3 in our manuscript, but (A) and (B) use the 0.1 coverage dataset excluding HGT genes, while (C) and (D) use the 0.5 coverage dataset excluding HGT genes.

Specific comments:

R2.6 • Line 130. For clarity it would be interesting to specify better in the introduction known patterns in filamentous ascomycetes, animals, and plants. In animals and plants some of the patterns are briefly mentioned at the beginning but in a rather confusing way (e.g. was it a gradual expansion/abrupt, accompanied by gene loss and on). Known patterns in filamentous ascomycetes are never mentioned. Please highlight better in the text if this is one of the first study for this specific comparison.

AUTHORS' RESPONSE: Thank you for your valuable feedback. We have revised the introduction to clearly specify the known patterns in filamentous ascomycetes as an example of novel fungal gene family expansions.

R2.7 • Line 156: could you better define the range of "equivalent"?

AUTHORS' RESPONSE: Thank you for your suggestion. We have revised the sentence to specify the gene count more precisely. The updated sentence now reads: “However, when comparing organisms with equivalent numbers of protein-coding genes (e.g., when comparing a yeast genome with ~10,000 genes with a plant genome with ~10,000 genes), yeasts displayed similar weighted average gene family sizes to plants and larger sizes than filamentous ascomycetes and animals (Figure 1B).”

R2.8 • Line 157: yeasts displayed similar weighted average sizes to plants and larger sizes than filamentous ascomycetes and animals (Figure 1b). I wonder if there is a better way to highlight this considering the overlapping points. Could you overlay yeast to plants and plants to yeast/further increase the alpha factor?

AUTHORS' RESPONSE: Thank you for your suggestions. We have updated Figure 1B by reordering the plot points to display yeasts above plants and adjusted the alpha transparency. These changes improve the visibility of overlapping points and clearly highlight that yeasts have similar weighted average sizes to plants and larger sizes than filamentous ascomycetes and animals.

R2.9 • Line 217. I am unable to find the panels 2k-m in Figure 2. If the absence is intentional please adjust the text.

AUTHORS' RESPONSE: Thank you for pointing this out. We have corrected the text to refer to Figure 3A instead of Figure 2.

R2.10 • Line 234. Figure legends and citations in the text might be inconsistent. First, please define better the 0.5 coverage threshold. Next, it feels like the 0.1 and 0.5 values go back and forth. In Figure S1 the threshold (red line) is at 0.1 but Figure S1 is cited for threshold 0.5. The main text Figure 3 has a PCA with coverage 0.1. Figure S6 on the other hand shows the threshold at 0.5 but the main text cites both figures as having 0.1. I would suggest putting the final threshold you used in the main text figure and the more relaxed one in a single supplementary figure. Justify better why 0.5 was used.
Also check which threshold was described in the methods Line 621.

AUTHORS' RESPONSE: Thank you for your detailed feedback. We have clarified the coverage thresholds by specifying that a 0.1 threshold is used for fold change analyses and a 0.5 threshold for PCA to ensure robust clustering. Figure S1 (Figure EV1) now displays the 0.1 and 0.5 threshold, while Figure S6 (Appendix Figure S3) shows the 0.1 threshold. Figure 3 in the main text exclusively uses the 0.5 threshold for PCA, and the Methods section has been updated to reflect these values consistently.

R2.11 • Line 253: PCA scatter plot (cite the specific panel you refer to).

AUTHORS' RESPONSE: We have clarified the citation of the PCA scatter plot by specifying that it refers to Figure 3C in the manuscript.

R2.12 • Line 620: connected to the previous point you mentioned in the methods citation 3 to support the use of PCA for the presence/absence analysis. The reference cited used a principal coordinate analysis (PCoA). Could you comment on the suitability of the 2 methods for your purpose? Also, a phylogenetic-aware analysis would have changed anything?

AUTHORS' RESPONSE: Thank you for your insightful comments. In order to compare the performance of PCA and Principal Coordinates Analysis (PCoA) in revealing the pattern of yeast gene family evolution, we employed PCoA using Euclidean, Manhattan, and Jaccard distance metrics on both presence/absence and copy number data. PCoA on presence/absence data, particularly with Euclidean and Jaccard distances, effectively separated FELs from SELs and enabled the identification of key gene families with enrichment results consistent with those obtained using PCA (Reviewer #2 Figures 4-7). Therefore, we retained the PCA results in the manuscript for consistency. Additionally, we explored phylogeny-aware PCA (phyPCA), which successfully differentiated FELs and SELs (Reviewer #2 Figure 8) but identified only 26 gene families, insufficient for robust enrichment analysis. Thus, PCA was determined to be the most suitable method for our study objectives.

Reviewer #2 Figure 4. PCoA on presence/absence dataset using Euclidean distance.

Reviewer #2 Figure 5. PCoA on presence/absence dataset using Manhattan distance.

Reviewer #2 Figure 6. PCoA on presence/absence dataset using Jaccard distance.

Reviewer #2 Figure 7. Functional enrichment analysis on PCoA1 using Jaccard and Euclidean results.

Reviewer #2 Figure 8. Phylogenetic PCA.

R2.13 • Line 318: in Figure 4d the difference in long and short dashes is not visible immediately.

AUTHORS' RESPONSE: We have revised Figure 4D to enhance the visibility of the long and short dashes by modifying the dash patterns and increasing their contrast.

R2.14 • Line 372: reference Figure 5.

AUTHORS' RESPONSE: We have updated Line 372 to include a reference to Figure 5.

R2.15 • Line 407: "specifically" is misplaced? Please rephrase the sentence for clarity.

AUTHORS' RESPONSE: We have revised the sentence by removing the word to enhance clarity.

R2.16 • Line 404: what are the potential implications in this last paragraph? Only a small portion of gains include novelty, and we are rather observing expansion of ancestral families?

AUTHORS' RESPONSE: Thank you for raising this important point. Although orphan gene families represent only a small fraction of gene gains, they play a crucial role in lineage-specific adaptations, driving evolutionary innovation and ecological specialization in yeast lineages through the introduction of unique metabolic functions and stress response mechanisms (Carvunis et al., 2012; Gaikani et al., 2024; Capra et al., 2010). To further investigate their evolutionary significance, we expanded our research to examine the origins of orphan gene families using BLASTP against the NCBI non-redundant (NR) protein database. Our analysis revealed that 96.5% of orphan gene families in yeast are likely of de novo origin. Detailed explanations of these findings have been added to the revised Results and Methods sections.

R2.17 • Line 417: the more specialised lifestyle/metabolic capacity is not so universal as this sentence hints to but has been identified specifically only in Dipodascales. When looking for more details regarding this aspect in other orders I can only find line 382 and a generic "gene families involved in metabolism". It would be interesting to expand, elaborate this part in order to corroborate the conclusion/statement in line 417.

AUTHORS' RESPONSE: Thank you for your valuable feedback. We have included detailed information in the Results section on the metabolic gene families that underwent frequent shifts. These additions further support and validate the statement: "marked by a transformation from a versatile SCA to descendants with more specialized lifestyles and metabolic capacities."

R2.18 • Line 527: could you annotate in e.g. supplementary table 1 which species were used as -core in orthofinder analysis?

AUTHORS' RESPONSE: Thank you for your suggestion. We have added annotations in Supplementary Table 1 to indicate which species were used as the “core” in the OrthoFinder analysis for clarity.

R2.19 • Line 544: can you specify the availability of this script or add it to the repository if not available yet? Same for the other R custom script in Line 632.

AUTHORS' RESPONSE: Thank you for bringing this to our attention. We have ensured that the script mentioned in Line 544 is now available in the repository, along with the R custom script mentioned in Line 632. Both scripts are accompanied by instructions to facilitate their use.

Reviewer #3:

R3.1 In this manuscript, the authors examined gene family evolution in the Saccharomycotina and find that these yeasts have smaller gene numbers and smaller weighted average gene family sizes than plants, animals and filamentous ascomycetes (Pezizomycotina), but that among animals, plants and Saccharomycotina with comparable gene numbers, the Saccharomycotina exhibit larger weighted average gene family sizes. Furthermore, faster-evolving Saccharomycotina lineages are found to exhibit higher speciation rates and higher rates of gene loss, with e.g. mRNA splicing and carbohydrate metabolism genes enriched among the lost genes. Overall, I think this an interesting study but there are a number of methodological aspects that I have questions about.

AUTHORS' RESPONSE: We thank you for your thoughtful summary of our manuscript and for highlighting the key findings of our study. We are pleased that you found our study on gene family evolution in Saccharomycotina interesting and recognized the key findings regarding gene numbers, family sizes, speciation rates, and gene loss patterns. We appreciate your questions regarding the methodological aspects and have implemented various approaches to address them. For the issue related to PCA, we have added a detailed section in the Methods section to prevent confusion.

Major comments

R3.2 - Half of the introduction is actually a summary of the main results of the analyses, using terms such as the weighted average gene family size that are only introduced later on in the results section. This summary overlaps substantially with other sections of the manuscript and should be shortened in my opinion.

AUTHORS' RESPONSE: Thank you for your feedback regarding the introduction. We agree that it currently includes a detailed summary of our main results, which can cause overlap with the later sections. In the revised manuscript, we have made the following changes:

1. Shortened the Introduction

We have removed or condensed the detailed discussion of our main findings. Instead, the Introduction now focuses on providing background context and outlining the knowledge gap that our study aims to address.

2. Reduced overlap with the results and discussion

By omitting the detailed analytical content from the Introduction, we have minimized redundancy across sections. The main findings and methodologies are now presented primarily in the Results and Methods sections, respectively.

We believe these revisions address your concern and improve the manuscript's organization. Thank you again for your constructive suggestion.

R3.3 - The weighted average gene family size measure is central to the analyses performed in the manuscript, but is not defined adequately in the results or methods sections (and also not in the 1K plant transcriptome paper where it has been used before). As I understand it, a weighted sum is made of the sizes of all gene families in some species, e.g. a Saccharomycotina yeast (although the description in the methods suggests the measure is calculated for a set of species as a whole), with the weight for a gene family being the inverse of the maximum copy number of that gene family in all Saccharomycotina. It is not clear why such weighting is necessary and what its effects are.

AUTHORS' RESPONSE: Thank you for highlighting the need to clarify the definition and rationale behind the weighted average gene family size measure. We have revised the Methods section to include a more detailed explanation, mathematical formulation, and rationale for its use, as outlined below:

1. Definition and calculation

In the revised Methods, we explicitly state that for each gene family i , we derive its maximum copy number across the Saccharomycotina clade, denoted as $max(copy_i)$. We then use the inverse of this maximum copy number, $w_i = 1/max(copy_i)$, as a weight for that gene family. Subsequently, for each species, the weighted average gene family size is computed as:

$$\text{Weighted Avg.Size} = \frac{\sum_{i=1}^n copy_i \times w_i}{n} \times \text{mean max}$$

This formula is now clearly presented in the revised manuscript so that readers can see precisely how the measure is computed.

2. Reason for weighting

We emphasize that the weighting strategy is designed to mitigate the influence of gene families that display large expansions in only one or a few species. In such cases, the total number of genes in these families can skew average-based metrics and overshadow changes in the rest of the gene families. By assigning a smaller weight to gene families with high maximum number, we reduce their disproportionate effect, allowing for a more balanced comparison across species.

3. Effects on the Analysis

Our analyses revealed that the trends observed using both unweighted and weighted metrics were largely consistent (Reviewer #3 Figure 1). This consistency suggests that, in our dataset, gene families with large numbers are not disproportionately influencing the overall patterns. Consequently, both unweighted and weighted approaches reliably reflect the underlying

evolutionary dynamics of gene families across species. The alignment of results from both methods reinforces the robustness of our conclusions, indicating that the observed patterns are inherent to the data rather than artifacts of the analytical method. Nonetheless, we maintain the use of the weighting strategy to safeguard against potential outliers or lineage-specific expansions, ensuring the reliability of our analysis in broader contexts.

We hope this additional clarification resolves concerns about the necessity and utility of the weighted average gene family size measure.

Reviewer #3 Figure 1. weighted average gene family size using weighting and unweighting methods. The arrangement of yeasts on the x-axis follows the same

order as in the phylogenetic tree and aligns with Figure S2 (Figure EV2) in our manuscript.

R3.4 The weighting scheme also relies heavily on the maximum gene family sizes in the set of species investigated, which are at risk of being outlier values due to e.g. suboptimal genome assemblies or annotations. It is also not clear to me why the resulting weighted averages would be comparable across different kingdoms, as species of different kingdoms would not be compared on equal footing because of the use of entirely different weighting schemes. In theory one badly assembled genome with inflated gene family sizes could be sufficient to skew the weighted averages of an entire kingdom.

AUTHORS' RESPONSE: We thank you for raising these important points about potential outliers and cross-kingdom comparability when using a weighted average approach. Below is how we addressed your questions:

1. Alternative weighting methods

We applied three additional weighting methods: (1) Excluding the top 5% of values and using the maximum value as the weight, (2) Using the mean value as the weight, and (3) Using the median value as the weight. We reran the primary analyses using these methods, and the overall patterns in yeast weighted average gene family size comparisons remained consistent with the original findings. FELs were clearly distinguishable from SELs (Reviewer #3 Figure 2). Similarly, comparisons across yeasts, filamentous ascomycetes, animals, and plants displayed consistent patterns, with strong correlations among yeasts, filamentous ascomycetes, and plants but weaker correlations in animals (Reviewer #3 Figure 3). Notably, yeasts exhibited the steepest slope for PICs of weighted average gene family size versus PICs of the total number of genes, suggesting yeasts tend to have larger gene family sizes at comparable gene levels.

2. Re-delineating gene families across different kingdoms

To ensure comparability across kingdoms, we reran OrthoFinder using all representative species of yeasts, filamentous ascomycetes, animals, and plants (as included in Figure 1A in our manuscript). We applied consistent filtering to exclude species-specific gene families with a coverage threshold of 0.25 and tested the three alternative weighting methods. The results were consistent with our original analysis (Reviewer #3 Figure 4), showing yeasts with relatively lower weighted average gene family sizes and plants forming two groups based on terrestrial and aquatic species.

Reviewer #3 Figure 2. weighted average gene family size using different weighting methods. The arrangement of yeasts on the x-axis follows the same order as in the phylogenetic tree and aligns with Figure S2 (Figure EV2) in our manuscript.

Reviewer #3 Figure 3. Comparison of weighted average gene family size across yeasts, filamentous ascomycetes, animals, and plants. Similar to Figure 1B and 1C in our manuscript, but generated using different weighting methods.

Reviewer #3 Figure 4. Comparison of weighted average gene family size across representative species of yeasts, filamentous ascomycetes, animals, and plants. Similar to Figure 1A in our manuscript, but generated using different weighting methods.

R3.5 - Related to this, it is unclear to me whether for the analyses across 12 yeast orders the weights have been calculated per order or for all orders simultaneously.

AUTHORS' RESPONSE: We thank you for highlighting this point. In our analysis, the weights are calculated across all 12 yeast orders simultaneously, rather than

separately for each order. We have updated the Methods section to clarify this approach and avoid any confusion for readers.

R3.6 - On several occasions, conflicting statements are made in different parts of the text. E.g. the abstract states that yeasts are 'characterized by small genome sizes and smaller gene numbers but larger gene family sizes' while in the introduction it is stated that (line 113-114) 'yeasts have smaller weighted average gene family sizes due to fewer gene counts'. This needs to be clarified: it is the ratio of average gene family size to gene number that is larger in Saccharomycotina than in other clades.

AUTHORS' RESPONSE: We thank you for pointing out these inconsistencies. We have clarified in the revised manuscript that what distinguishes Saccharomycotina is not having “larger gene family sizes”, but rather a higher ratio of average gene family size relative to their total gene count. We have updated the relevant statements in both the abstract and the introduction to avoid any contradictory wording.

R3.7 - Similarly, the authors state on line 433 that 'These gene family contractions and losses in FELs may contribute to their higher evolutionary and speciation rates (Figures 2e-f and 3e) by enabling rapid genomic adaptations that optimize cellular processes crucial for survival and reproduction in diverse and challenging environments.', while on line 467 it is stated that 'when we control for gene number, we found that yeasts exhibit larger weighted average gene family sizes than both filamentous ascomycetes and animals and are on par with plants (Figure 1b). Larger gene family sizes may provide redundancy and adaptability in biological processes'. These statements seem to be at odds with each other. While gene losses may indeed conceivably lead to higher speciation rates if there is differential gene loss in different subpopulations leading to reproductive isolation, I don't see the link between increased gene loss and increased survival or adaptability.

AUTHORS' RESPONSE: Thank you for your thoughtful feedback regarding the interpretation of our results. We appreciate the opportunity to clarify these concepts and have revised the manuscript to address this issue more cohesively.

The first statement refers to fast-evolving lineages (FELs) within yeasts, where gene family contractions and losses may facilitate rapid genomic adaptations to diverse and challenging environments, potentially contributing to higher evolutionary and speciation rates. This is a lineage-specific observation within yeasts. In contrast, the second statement pertains to a broader inter-kingdom comparison, where we observe that yeasts, as a group, have larger weighted average gene family sizes compared to filamentous ascomycetes and animals, and are on par with plants. This highlights the evolutionary potential of yeasts as a whole, rather than focusing on specific lineages.

We provided a comprehensive discussion on the relationship between gene loss and species' adaptive abilities. Additionally, we examined the mechanisms of gene loss and its potential impacts, emphasizing that gene loss can have adaptive, neutral, or deleterious effects depending on the context. We hope these revisions resolve the apparent contradiction and better communicate the complementary roles of gene expansions and losses in shaping yeast evolution. Thank you for your insightful comments, which have strengthened our Discussion.

R3.8 - The authors base their gene family expansion and contraction statements in FEL versus SEL yeasts on fold changes in gene family size, equating e.g. fold changes >1.5 in FEL versus SEL yeasts as expansions in the FEL lineage. Couldn't these equally likely be contractions in the SEL lineage instead? Why didn't the authors use e.g. CAFE here to assess expansions and contractions, as they do further on in the manuscript? Without reference to gene family sizes in the most recent common ancestor, any assessment of expansion or contraction in a particular lineage seems arbitrary.

AUTHORS' RESPONSE: We thank you for highlighting this concern. You raise an important point about the assumption of an ancestral reference state in fold change analysis and the potential for ambiguity in labeling expansions versus contractions. We addressed your points as follows:

1. General patterns in FELs vs. outgroup clades
Compared to their relative outgroup clades (e.g., Phaffomycetales, an outgroup to Saccharomycodales in Figure S2 (Figure EV2)), FELs showed significantly lower weighted average gene family sizes, indicating widespread contraction/loss events in these lineages (Figure S2 (Figure EV2) in our manuscript).
2. Comparison with the MRCA of their respective orders
Interpreting CAFE results is challenging with large phylogenetic trees due to the numerous ancestral branches. For example, a tree with 50 species has 99 ancestral branches, making functional enrichment and pattern analysis difficult. Additionally, CAFE is typically limited to handling phylogenies with up to approximately 200 species. In our study, we used CAFE (Figure 5 in our manuscript) to investigate general patterns within the Saccharomycotina common ancestor (SCA) lineages by focusing on 12 orders, which reduced the number of primary branches to 23. Within three of these orders, comprising 15, 184, and 24 species respectively, we conducted fold change analyses to ensure consistent comparisons across all three groups. If we focus solely on extant species, the fold change comparison described in our manuscript suffices. However, to verify that these significantly changed gene families also followed similar evolutionary trends when compared to ancestral yeasts, we analyzed CAFE results for ancestral yeast gene family

copy numbers. Specifically, we compared FEL averages to the MRCA (most recent common ancestor) of their respective orders. This analysis targeted gene families that showed significant expansion/gain or contraction/loss in the fold change analysis. Notably, across all three orders, most gene families exhibited similar trends (Reviewer #3 Figure 5B).

To clarify these points and prevent misunderstanding, we have revised the Methods section and included a new figure to distinguish between a direct FEL vs. SEL comparison and a phylogenetic approach referencing ancestral states.

Reviewer #3 Figure 5. Average gene family copy number comparisons. This figure presents comparisons of average gene family copy numbers, focusing only on gene families identified as experiencing significant changes in the fold change analysis. **(A)** Average gene family copy numbers of FELs compared to their respective SELs. The solid diagonal line represents equal values; points below the

line indicate that FELs have larger average numbers than SELs. In the contraction/loss panels, annotations such as “534/536” and “13/13” indicate that 534 out of 536 and 13 out of 13 gene families, respectively, in FELs are smaller than in SELs. **(B)** Average gene family copy numbers of FELs compared to the MRCA of their respective orders.

R3.9 - The authors use PCA analysis on binary presence-absence data for gene families and afterwards identify the gene families that contributed most to the first two PCs. Using PCA on binary data is however not advised for anything else than pure dimensionality reduction (i.e. not for interpretation of variable loadings on the components), see e.g. Collins, M., Dasgupta, S., & Schapire, R. E. (2002). A generalization of principal components analysis to the exponential family. *Advances in neural information processing systems* pp. 617-624. Using PCA on the full gene family size data rather than presence-absence would have been better although still not ideal.

AUTHORS' RESPONSE: We appreciate your insights and the reference to Collins et al. (2002). We recognize the limitations of using PCA on binary presence-absence data for interpretation and have taken the following steps to address these limitations:

1. PCA on gene family number data

We performed PCA on gene family number datasets using scaled and unscaled methods. However, this approach showed lower discriminatory power for distinguishing yeasts and FELs/SELs compared to PCA on presence/absence data (Figure 3C in our manuscript and Reviewer #3 Figure 6).

2. PCoA on presence/absence and gene family number data

We applied PCoA with multiple distance metrics (Euclidean, Manhattan, Jaccard) to both datasets. PCoA performed better on presence/absence data, with Jaccard and Euclidean metrics yielding the best results (Reviewer #3 Figures 7-9). Using this approach, we identified 164 and 373 gene families contributing to yeast distribution differences along PCoA1 for Jaccard and Euclidean metrics, respectively. These gene families were enriched for functions like the respiratory chain complex I and mitochondrial respirasome, but not the DASH complex analyzed in later sections (Figures 3D, 4A, and 4C in our manuscript).

While PCA on binary data has inherent limitations, it was the most effective method we tested (compared to PCoA, t-SNE, and UMAP) for distinguishing FELs and SELs. Our primary goal is to identify yeast genomes or lineages with significant changes in gene content and the major gene families driving these changes. Despite theoretical limitations, our findings are supported by further validation (Figure 4 in our

manuscript) and align with results from alternative approaches. We have updated the Methods section to clarify our rationale and discuss these potential limitations.

Reviewer #3 Figure 6. PCA on copy number datasets.

Reviewer #3 Figure 7. PCoA on presence/absence dataset using Euclidean distance.

Reviewer #3 Figure 8. PCoA on presence/absence dataset using Manhattan distance.

Reviewer #3 Figure 9. PCoA on presence/absence dataset using Jaccard distance.

Reviewer #3 Figure 10. Functional enrichment analysis on PCoA1 using Jaccard and Euclidean results.

Minor comments

R3.10 - Line 102 and other instances : 'yeast' is not synonymous to 'Saccharomycotina'

AUTHORS' RESPONSE: Thank you for pointing this out. We have added a clarification in Line 102 to specify that, in this manuscript, “yeast” is used to refer to “Saccharomycotina.”

R3.11 - Line 212: '... for each gene family and each pair': it is not clear what the pairs are here.

AUTHORS' RESPONSE: Thank you for pointing this out. To clarify, the “each pair” mentioned in Line 212 refers to the FEL and SEL within the same order. We have revised the text to make this distinction clear.

R3.12 - Line 646: 'species with larger coordinates on the PC tend to have more gene copies in the top *i* gene families' : I don't see why this should be the case, as the number of gene copies was not taken into account in the PCA analysis.

AUTHORS' RESPONSE: Thank you for pointing this out. To clarify, we calculated the correlation between the PC coordinates and the average copy numbers of the top *i* gene families. This approach allows us to evaluate whether species with larger PC coordinates tend to have higher average gene copy numbers in these families. We have revised the text to make this methodology clearer.

5th Apr 2025

Manuscript Number: MSB-2024-12697R

Title: Unique trajectory of gene family evolution from genomic analysis of nearly all known species in an ancient yeast lineage

Author: Bo Feng

Yonglin Li

Biyang Xu

Hongyue Liu

Jacob Steenwyk

Kyle David

Xiaolin Tian

Carla Goncalves

Dana Opulente

Abigail LaBella

Marie-Claire Harrison

John Wolters

Shengyuan Shao

Zhaohao Chen

Kaitlin Fisher

Marizeth Groenewald

Chris Hittinger

Xing-Xing Shen

Shengying Li

Antonis Rokas

Xiaofan Zhou

Yuanning Li

Dear Prof Rokas,

Thank you again for submitting your work to Molecular Systems Biology. We have now heard back from the three original referees who agreed to re-evaluate the revised study. As you will see, the referees are highly supportive and find the work valuable and important. Yet, Reviewer #3 raises a series of comments and make suggestions for minor modifications, which we would ask you to address in a minor revision of the present work.

Please resubmit your revised manuscript online, with a covering letter listing amendments and responses to each point raised by the referee. Please resubmit the paper ****within one month**** and ideally as soon as possible.

When you resubmit your manuscript, please download our CHECKLIST (<https://bit.ly/EMBOPressAuthorChecklist>) and include the completed form in your submission. *Please note* that the Author Checklist will be published alongside the paper as part of the transparent process (<https://www.embopress.org/page/journal/17444292/authorguide#transparentprocess>)

Click on the link below to submit your revised paper.

Yours sincerely,

Yehu Moran

Academic Editor

Molecular Systems Biology

When you are ready to resubmit, please click on the link below to submit the revision online before 5th May 2025.

IMPORTANT: When you send your revision, we will require the following items:

1. the manuscript text in LaTeX, RTF or MS Word format
2. a letter with a detailed description of the changes made in response to the referees. Please specify clearly the exact places in the text (pages and paragraphs) where each change has been made in response to each specific comment given
3. three to four 'bullet points' highlighting the main findings of your study
4. a short 'blurb' text summarizing in two sentences the study (max. 250 characters)
5. a 'thumbnail image' (550px width and max 400px height, Illustrator, PowerPoint or jpeg format), which can be used as 'visual title' for the synopsis section of your paper.
6. Please include an author contributions statement after the Acknowledgements section (see <https://www.embopress.org/page/journal/17444292/authorguide#manuscriptpreparation>)
7. Please complete the CHECKLIST available at (<https://bit.ly/EMBOPressAuthorChecklist>). Please note that the Author Checklist will be published alongside the paper as part of the transparent process (<https://www.embopress.org/page/journal/17444292/authorguide#transparentprocess>).
8. When assembling figures, please refer to our figure preparation guideline in order to ensure proper formatting and readability in print as well as on screen:
<https://bit.ly/EMBOPressFigurePreparationGuideline>
See also figure legend guidelines: <https://www.embopress.org/page/journal/17444292/authorguide#figureformat>
9. Please note that corresponding authors are required to supply an ORCID ID for their name upon submission of a revised manuscript (EMBO Press signed a joint statement to encourage ORCID adoption). (<https://www.embopress.org/page/journal/17444292/authorguide#editorialprocess>)
Currently, our records indicate that the ORCID for your account is 0000-0002-7248-6551.

Link Not Available

10. Include a Reagents and Tools Table as part of the Methods section, which can be downloaded from our author guidelines (<https://www.embopress.org/page/journal/17444292/authorguide#structuredmethods>)

*** PLEASE NOTE *** As part of the EMBO Press transparent editorial process initiative (see our Editorial at <https://dx.doi.org/10.1038/msb.2010.72> , Molecular Systems Biology will publish online a Review Process File to accompany accepted manuscripts. When preparing your letter of response, please be aware that in the event of acceptance, your cover letter/point-by-point document will be included as part of this File, which will be available to the scientific community. More information about this initiative is available in our Instructions to Authors. If you have any questions about this initiative, please contact the editorial office (msb@embo.org).

Reviewer #1:

I am satisfied that the authors have appropriately addressed all requests/concerns from the reviewers. They have undertaken substantial amount of work which has improved the overall quality and flow of the manuscript. I am happy to see this manuscript published and commend the authors for this work.

Reviewer #2:

I would like to thank the authors for their attention to all reviewers' comments. All my concerns have been addressed and I believe the manuscript has been greatly improved. In my opinion it is now suitable for publication in MSB. Congratulations!

Reviewer #3:

I thank the authors for the extensive extra analyses they performed in response to my previous comments, and I appreciate the extended discussion in the revised manuscript. I have no further major comments. My residual minor comments are the following :

- Regarding the following statement that was added to the Methods : 'We also explored alternative weighting methods, including: (1) excluding the top 5% of values and using the next highest value as the weight, (2) using the mean value as the weight, and (3) using the median value as the weight. The primary analyses were repeated with these methods, and the overall patterns in yeast weighted average gene family size comparisons remained consistent with the original results.', I think the authors should add the results figures using these alternatives (as presented in the response to reviewer #3) to the supplementary data, so that readers can check them out.
- Idem for the added statement : 'To verify that the gene families identified as significantly changed (expansions/gains or contractions/losses) in the fold change analysis exhibited similar evolutionary trends relative to ancestral yeasts, we examined their ancestral gene family copy numbers using CAFE results. Specifically, we compared the average copy numbers of FELs to those of the MRCA for each order. Across all three orders, most gene families exhibited similar evolutionary patterns.'. These analyses should be added to the supplementary data.
- Idem for the added statement : 'To assess the robustness of our methods for detecting yeast genomes/lineages with significant changes in gene content and identifying the major gene families responsible, we performed PCA on the copy number data matrix and PCoA on both the presence/absence and copy number data. Both PCA and PCoA performed better on the presence/absence data, and they showed similar patterns in identifying key gene families that explain the distribution of yeasts on PCA/PCoA scatter plots. Although PCA has theoretical limitations when applied to binary (presence/absence) data, our findings are validated by further evidence (Fig. 4) and align with results from alternative approaches.' Readers should be able to compare these analyses by themselves.
- Idem for the results obtained when excluding HGT-related genes generated in response to a comment of Reviewer #2. Adding all the results generated in response to the reviewers' comments as supplementary data/figures will strengthen the manuscript.
- Regarding Reviewer #3 Figure 5, panel A : am I correct in assuming the dots shown in e.g. the contraction panel for Dipodascales are the gene families identified as contracted by the fold change analysis ? If so, how can some of the points be located below or close to the diagonal when the fold change cutoff FEL/SEL = 0.67 ?
- Figure 2A in the track-changes version is slightly different from Figure 2A in the merged PDF version of the revision. In the merged PDF, colors for Pichiales and Alaninales on the phylogeny have been switched relative to the colors of the violin plots.

Response to Reviewers

Reviewers' Comments

Reviewer #1:

R1.1 I am satisfied that the authors have appropriately addressed all requests/concerns from the reviewers. They have undertaken substantial amount of work which has improved the overall quality and flow of the manuscript. I am happy to see this manuscript published and commend the authors for this work.

AUTHORS' RESPONSE: We have greatly appreciated your thorough evaluation and have taken the necessary steps to address all the reviewers' concerns. We are delighted to hear that our revisions have improved the manuscript's quality and flow, and we thank you for your recommendation to publish our work.

Reviewer #2:

R2.1 I would like to thank the authors for their attention to all reviewers' comments. All my concerns have been addressed and I believe the manuscript has been greatly improved. In my opinion it is now suitable for publication in MSB. Congratulations!

AUTHORS' RESPONSE: Thank you for recognizing our efforts to address all the reviewers' suggestions. We are delighted to know that you find our manuscript greatly improved and suitable for publication. We truly appreciate your support.

Reviewer #3:

R3.1 I thank the authors for the extensive extra analyses they performed in response to my previous comments, and I appreciate the extended discussion in the revised manuscript. I have no further major comments. My residual minor comments are the following:

AUTHORS' RESPONSE: Thank you for your careful reevaluation and for acknowledging our additional analyses and extended discussion. We have

carefully addressed all of your residual minor comments and have integrated the suggested revisions into the manuscript. We appreciate your time and effort in reviewing our work.

R3.2 - Regarding the following statement that was added to the Methods: 'We also explored alternative weighting methods, including: (1) excluding the top 5% of values and using the next highest value as the weight, (2) using the mean value as the weight, and (3) using the median value as the weight. The primary analyses were repeated with these methods, and the overall patterns in yeast weighted average gene family size comparisons remained consistent with the original results.', I think the authors should add the results figures using these alternatives (as presented in the response to reviewer #3) to the supplementary data, so that readers can check them out.

AUTHORS' RESPONSE: Thank you for the suggestion. We have added the requested figures illustrating the alternative weighting methods to the supplementary data, as recommended, so readers can verify these results for themselves.

R3.3 - Idem for the added statement: 'To verify that the gene families identified as significantly changed (expansions/gains or contractions/losses) in the fold change analysis exhibited similar evolutionary trends relative to ancestral yeasts, we examined their ancestral gene family copy numbers using CAFE results. Specifically, we compared the average copy numbers of FELs to those of the MRCA for each order. Across all three orders, most gene families exhibited similar evolutionary patterns.'. These analyses should be added to the supplementary data.

AUTHORS' RESPONSE: Thank you for your feedback. We have added these analyses to the supplementary data, so that readers can examine the evolutionary patterns of significantly changed gene families in comparison to their respective ancestral states.

R3.4 - Idem for the added statement: 'To assess the robustness of our methods for detecting yeast genomes/lineages with significant changes in gene content and identifying the major gene families responsible, we performed PCA on the copy number data matrix and PCoA on both the presence/absence and copy number data. Both PCA and PCoA performed better on the presence/absence data, and they showed similar patterns in

identifying key gene families that explain the distribution of yeasts on PCA/PCoA scatter plots. Although PCA has theoretical limitations when applied to binary (presence/absence) data, our findings are validated by further evidence (Fig. 4) and align with results from alternative approaches.' Readers should be able to compare these analyses by themselves.

AUTHORS' RESPONSE: Thank you for suggesting this addition. We have included these PCA/PCoA results in the supplementary data so that readers can directly compare these analyses and verify the key gene families influencing the distribution of yeasts on the scatter plots.

R3.5 - Idem for the results obtained when excluding HGT-related genes generated in response to a comment of Reviewer #2. Adding all the results generated in response to the reviewers' comments as supplementary data/figures will strengthen the manuscript.

AUTHORS' RESPONSE: Thank you for the suggestion. We have added the results obtained when excluding HGT-related genes to the supplementary data/figures, to ensure full transparency and to strengthen the overall manuscript.

R3.6 - Regarding Reviewer #3 Figure 5, panel A: am I correct in assuming the dots shown in e.g. the contraction panel for Dipodascales are the gene families identified as contracted by the fold change analysis? If so, how can some of the points be located below or close to the diagonal when the fold change cutoff $FEL/SEL = 0.67$?

AUTHORS' RESPONSE: Thank you for your question. Yes, those points in Reviewer #3 Figure 5 panel A represent the gene families identified as contracted using our fold change cutoff of 0.67. The apparent discrepancy arises because, in Figure 5 panel A, we initially plotted the direct average copy numbers (including all data points), whereas for the fold change calculation, we excluded the top and bottom 5% of outliers. This difference in data handling led to slight variations in the plotted values, causing some points to appear below or close to the diagonal. We have now revised and recalculated the figure using the same method as in the fold change analysis (Reviewer #3 Figure 1).

Reviewer #3 Figure 1. Average gene family copy number comparisons.

This figure presents comparisons of average gene family copy numbers, focusing only on gene families identified as experiencing significant changes in the fold change analysis. **(A)** Average gene family copy numbers of FELs compared to their respective SELs. The solid diagonal line represents equal values; points below the line indicate that FELs have larger average numbers than SELs. In the contraction/loss panels, annotations such as “536/536” and “13/13” indicate that 536 out of 536 and 13 out of 13 gene families, respectively, in FELs are smaller than in SELs. **(B)** Average gene family copy numbers of FELs compared to the MRCA of their respective orders.

R3.7 - Figure 2A in the track-changes version is slightly different from Figure 2A in the merged PDF version of the revision. In the merged PDF, colors for Pichiales and Alaninales on the phylogeny have been switched relative to the colors of the violin plots.

AUTHORS' RESPONSE: Thank you for pointing this out. We have corrected the color mismatch between the two versions of Figure 2A so that both the phylogeny and the violin plots are now aligned consistently.

30th Apr 2025

Manuscript number: MSB-2024-12697RR

Title: Unique trajectory of gene family evolution from genomic analysis of nearly all known species in an ancient yeast lineage

Dear Prof Rokas,

Thank you again for sending us your revised manuscript. We are now satisfied with the modifications made and I am pleased to inform you that your paper has been accepted for publication.

Yours sincerely,

Sincerely,

Yehu Moran
Academic Editor
Molecular Systems Biology
